# Improving statistical power in severe malaria genetic association studies by augmenting phenotypic precision

James A Watson[1,2†*], Carolyne M Ndila[1,2†], Sophie Uyoga[3], Alexander Macharia[3], Gideon Nyutu[3], Shebe Mohammed[3], Caroline Ngetsa[3], Neema Mturi[3], Norbert Peshu[3], Benjamin Tsofa[3], Kirk Rockett[4,5], Stije Leopold[1,2], Hugh Kingston[1,2], Elizabeth C George[6], Kathryn Maitland[3,7], Nicholas PJ Day[1,2], Arjen M Dondorp[1,2], Philip Bejon[2,3], Thomas N Williams[3,7‡], Chris C Holmes[8,9‡], Nicholas J White[1,2‡]

[1]Mahidol Oxford Tropical Medicine Research Unit, Faculty of Tropical Medicine, Mahidol University, Bangkok, Thailand; [2]Centre for Tropical Medicine and Global Health, Nuffield Department of Medicine, University of Oxford, Oxford, United Kingdom; [3]KEMRI-Wellcome Trust Research Programme, Centre for Geographic Medicine Research-Coast, Kilifi, Kenya; [4]The Wellcome Sanger Institute, Cambridge, United Kingdom; [5]Wellcome Trust Centre for Human Genetics, University of Oxford, Oxford, United Kingdom; [6]Medical Research Council Clinical Trials Unit, University College London, London, United Kingdom; [7]Institute of Global Health Innovation, Imperial College, London, London, United Kingdom; [8]Nuffield Department of Medicine, University of Oxford, Oxford, United Kingdom; [9]Department of Statistics, University of Oxford, Oxford, United Kingdom

*For correspondence:
jwatowatson@gmail.com

[†]These authors contributed equally to this work
[‡]These authors also contributed equally to this work

Competing interests: The authors declare that no competing interests exist.

**Abstract** Severe falciparum malaria has substantially affected human evolution. Genetic association studies of patients with clinically defined severe malaria and matched population controls have helped characterise human genetic susceptibility to severe malaria, but phenotypic imprecision compromises discovered associations. In areas of high malaria transmission, the diagnosis of severe malaria in young children and, in particular, the distinction from bacterial sepsis are imprecise. We developed a probabilistic diagnostic model of severe malaria using platelet and white count data. Under this model, we re-analysed clinical and genetic data from 2220 Kenyan children with clinically defined severe malaria and 3940 population controls, adjusting for phenotype mis-labelling. Our model, validated by the distribution of sickle trait, estimated that approximately one-third of cases did not have severe malaria. We propose a data-tilting approach for case-control studies with phenotype mis-labelling and show that this reduces false discovery rates and improves statistical power in genome-wide association studies.

## Introduction

Severe malaria caused by the parasite *Plasmodium falciparum* kills nearly half a million children each year, mostly in sub-Saharan Africa (*World Health Organization, 2020*). By causing death in children before they reach their reproductive age, *P. falciparum* has exerted a substantial selective evolutionary pressure on the human genome (*Carter and Mendis, 2002*; *Kariuki and Williams, 2020*). Recent advances in whole-genome sequencing and haplotype imputation (*Teo et al., 2010*), combined with data gathered prospectively from large patient cohorts, have improved our understanding of genetic susceptibility to *P. falciparum* infection and severe disease (*Malaria Genomic Epidemiology*

**eLife digest** In areas of sub-Saharan Africa where malaria is common, most people are frequently exposed to the bites of mosquitoes carrying malaria parasites, so they often have malaria parasites in their blood. Young children, who have not yet built up strong immunity against malaria, often fall ill with severe malaria, a life-threatening disease. It is unclear why some children develop severe malaria and die, while other children with high numbers of parasites in their blood do not develop any apparent symptoms.

Genetic susceptibility studies are designed to uncover why such differences exist by comparing individuals with severe malaria (referred to as 'cases') with individuals drawn from the general population (known as 'controls'). But severe malaria can be a challenge to diagnose. Since high numbers of malaria parasites can be found in healthy children, it is sometimes difficult to determine whether the parasites are making a child ill, or whether they are a coincidental finding. Consequently, some of the 'cases' recruited into these studies may actually have a different disease, such as bacterial sepsis. This ultimately affects how the studies are interpreted, and introduces error and inaccuracy into the data.

Watson, Ndila et al. investigated whether measuring blood biomarkers in patients (derived from the complete blood count, including platelet counts and white blood cell counts) could improve the accuracy with which malaria is diagnosed. They developed a new mathematical model that incorporates platelet and white blood cell counts. This model estimates that in a large cohort of 2,220 Kenyan children diagnosed with severe malaria, around one third of enrolled children did not actually have this disease. Further analysis suggests that patients with severe malaria are highly unlikely to have platelet counts higher than 200,000 per microlitre. This defines a cut-off that researchers can use to avoid recruiting patients who do not have severe malaria in future studies. Additionally, the ability to diagnose severe malaria more accurately can make it easier to detect and treat other diseases with similar symptoms in children with high numbers of malaria parasites in their blood.

Watson, Ndila et al.'s findings support the recommendation that all children with suspected malaria be given broad spectrum antibiotics, as many misdiagnosed children will likely have bacterial sepsis. It also suggests that using complete blood counts, which are cheap to obtain and increasingly available in low-resource settings, could improve diagnostic accuracy in future clinical studies of severe malaria. This could ultimately improve the ability of these studies to find new treatments for this life-threatening disease.

*Network et al., 2013*; *Malaria Genomic Epidemiology Network, 2014*; *Band et al., 2019*; *Malaria Genomic Epidemiology Network et al., 2017*), but many questions remain unanswered (*Kariuki and Williams, 2020*). A major limitation of genetic association studies in severe malaria is that the diagnosis of severe falciparum malaria in children is imprecise (*White et al., 2013*; *Taylor et al., 2004*; *Bejon et al., 2007*). This imprecision increases with transmission intensity because of the low positive predictive value of a 'positive blood film' or rapid diagnostic test (RDT) in areas where the background prevalence of microscopy detectable parasitaemia in apparently healthy young children is high (often around 30%, *Rodriguez-Barraquer et al., 2018*, but can exceed 90%, *Smith et al., 1994*).

Severe falciparum malaria has been defined by experts convened by the World Health Organization (WHO) as clinical or laboratory evidence of vital organ dysfunction in the presence of circulating asexual *P. falciparum* parasitaemia (*World Health Organisation, 2014*). The WHO definition of severe malaria is aimed primarily at clinicians and health care workers managing patients with malaria who appear severely ill. This appropriately prioritises sensitivity over specificity (*Anstey and Price, 2007*). An inclusive clinical definition ensures that cases are not missed and patients receive the best treatment. In contrast, genetic association studies require high specificity (*Zondervan and Cardon, 2007*). For a given sample size, their statistical power, false discovery rates (FDRs) and the validity of their interpretation are weakened by phenotypic inaccuracy. Specificity in the diagnosis of severe malaria depends in part on the prevalence of malaria parasitaemia. This reflects background transmission intensity. In areas of low or seasonal transmission (e.g. most of endemic Asia and the

Americas), clinical and laboratory signs of severity accompanied by a positive blood film for *P. falciparum* are highly specific for severe malaria, which predominantly affects young adults. In contrast in high transmission areas in sub-Saharan Africa and in lowland areas of the island of New Guinea, where severe malaria is largely a disease of young children, the diagnostic criteria for defining severe malaria are less specific because of the high background prevalence of asymptomatic parasitaemia and the lower specificity of the clinical manifestations. Standard case definitions of severe malaria will therefore inevitably include both patients with non-malarial severe illness with concomitant parasitaemia and with concomitant non-severe malaria.

Our goal was to develop a biomarker-based model that can differentiate probabilistically between 'true severe malaria' and severe illness not caused primarily by malaria, but with concomitant parasitaemia. We define 'true severe malaria' conceptually as a febrile illness caused by malaria parasites, with organ dysfunction, that can result in death whereby mortality is attributable directly to the malaria parasites. This attributable mortality can be given a formal causal definition by using a conceptual (albeit unethical) randomised experiment of delayed versus prompt antimalarial therapy. In a theoretical patient population with true severe malaria, delay in administration of an effective antimalarial would result in increased mortality (*Warrell et al., 1982*; *Gomes et al., 2009*) whereas in a population with severe illness not caused by malaria ('not severe malaria') there would not be a corresponding increase in mortality.

We developed a probabilistic diagnostic model of severe malaria based on haematological biomarkers using data from 1704 adults and children mainly from low transmission settings whose diagnosis of severe malaria is considered to be highly specific. We used this model to demonstrate low phenotypic specificity in a cohort of 2220 Kenyan children who were diagnosed clinically with severe malaria. We validated the predictions using a natural experiment, the distribution of sickle cell trait (HbAS), the genetic polymorphism with the strongest known protective effect against all forms of clinical malaria (*Malaria Genomic Epidemiology Network, 2014*). Building on work on 'data-tilting' (*Nie et al., 2013*), we suggest a new method for testing genetic associations in the context of case-control studies in which cases are re-weighted by the probability that the severe malaria diagnosis is correct under the model. As proof of concept, we ran a genome-wide association study across 9.6 million imputed biallelic variants using the subset of cases with genome-wide genotype data (n = 1297) and population controls (n = 1614). Adjusting for case mis-classification decreased genome-wide FDRs (*Storey, 2002*) and increased effect sizes in three of the top regions of the human genome most strongly associated with protection from severe malaria in East Africa (*HBB*, *ABO* and *FREM3*, *Band et al., 2019*). A re-analysis of 120 directly typed polymorphisms in 70 candidate malaria-protective genes in the 2220 Kenyan cases and 3940 population controls, examining differential effects between correctly and incorrectly classified cases, suggests that the protective effect of glucose-6-phosphate dehydrogenase (G6PD) deficiency has been obscured in this population by case mis-classification. Our results show that adding full blood count metadata – routinely measured in most hospitals in sub-Saharan Africa – to severe malaria cohorts would lead to more accurate quantitative analyses in case-control studies and increased statistical power.

## Results

### Reference model of severe malaria

We used the joint distribution of platelet counts and white blood cell counts (both on a logarithmic scale) to develop a simple biomarker-based reference model of severe malaria. To fit the reference model (i.e. P[Data | Severe malaria]), we used platelet and white count data from (i) severe malaria patient cohorts enrolled in low transmission areas where severe disease accompanied by a positive blood stage parasitaemia has a high positive predictive value for severe malaria (930 adults from Vietnam [*Hien et al., 1996*; *Phu et al., 2010*] and 653 adults and children from Thailand and Bangladesh); and (ii) severely ill African children with plasma *Pf*HRP2 concentrations >1000 ng/mL and >1000 parasites per μL of blood (121 children from Uganda, *Maitland et al., 2011*). Severe illness accompanied by a high plasma *Pf*HRP2 concentration makes the diagnosis of severe falciparum malaria highly specific (*Hendriksen et al., 2012*). The joint distribution of platelet and white blood cell counts in severe malaria was modelled as a bivariate *t*-distribution with both blood count variables on the $log_{10}$ scale.

*Figure 1A* shows the reference data (green triangles: patients with a highly specific diagnosis of severe malaria, summarised in *Table 1*) alongside data from a large Kenyan cohort of hospitalised children diagnosed with severe malaria, whose diagnosis had unknown specificity (pink squares). The median platelet count in the reference data was 57,000 per μL, and the median total white blood cell count was 8400 per μL. In contrast, the median platelet count in the Kenyan children was 120,000 per μL, and the median total white blood cell count was 13,000 per μL. Direct comparisons of white counts across these two datasets are confounded by geography and age. Total white blood cell counts are known to be age-dependent and vary across genetic backgrounds, in particular lower neutrophil counts are associated with mutations in the *ACKR1* gene that results in the Duffy negative phenotype prevalent in African populations (*Reich et al., 2009*). However, after adjustment for age (see Materials and methods), the marginal distributions of total white counts were comparable between Asian adults and children with severe malaria and African children with high plasma *Pf*HRP2 (Appendix 1). Platelet counts are not age-dependent and do not vary substantially across genetic backgrounds. The marginal distributions of platelet counts were comparable between Asian adults and children with severe malaria and African children with high plasma *Pf*HRP2 (Appendix 2). A low platelet count (thrombocytopenia) is a universal feature of severe malaria (see evidence collated in Materials and methods). To illustrate this important point, in a cohort of 566 severely ill Ugandan children enrolled in the Fluid Expansion as Supportive Therapy (FEAST) trial (*Maitland et al., 2011*), a trial including all severe illness not restricted to severe malaria, low platelet counts were highly

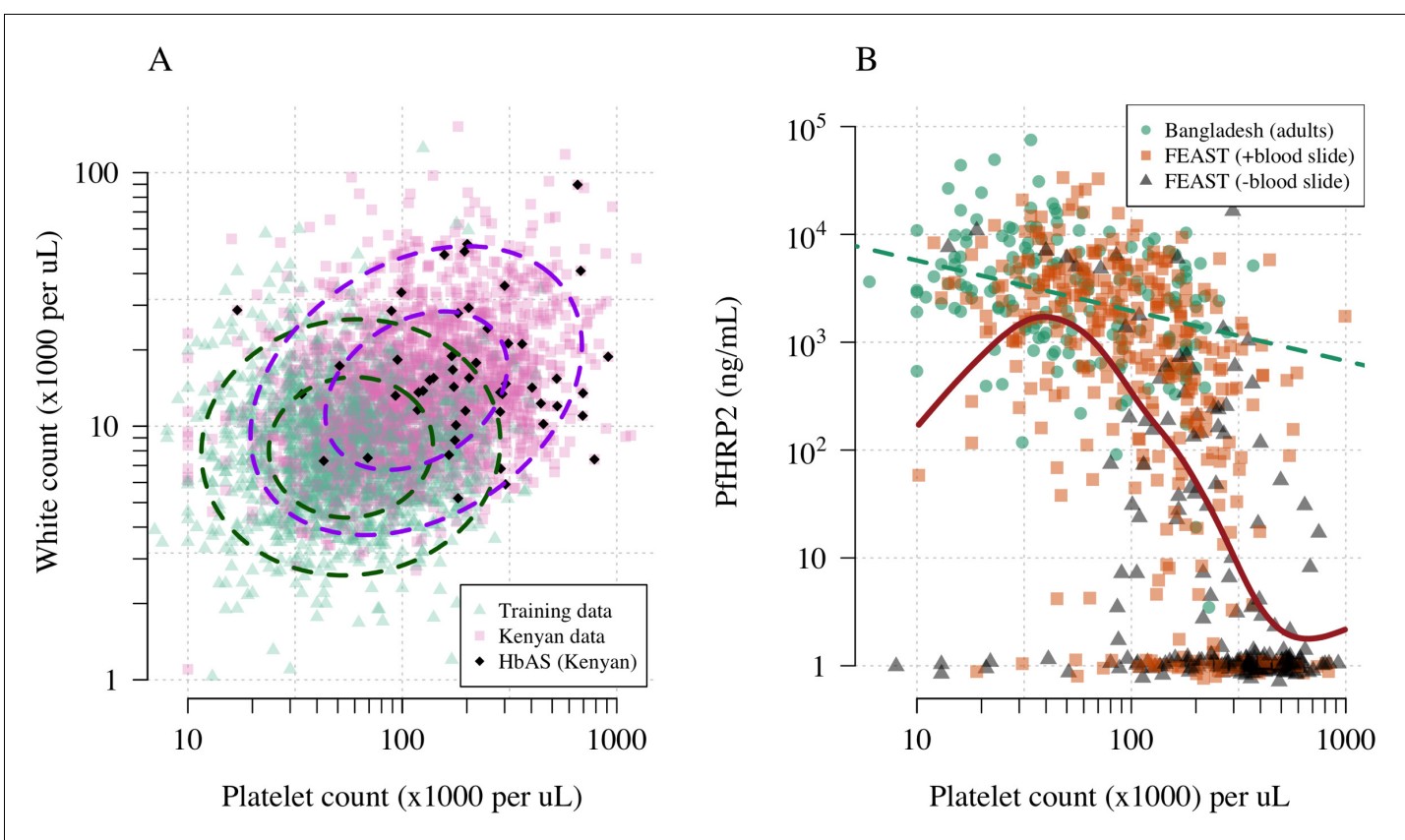

**Figure 1.** Platelet counts and white blood cell counts as diagnostic predictors of severe falciparum malaria. Panel (**A**) shows the bivariate marginal distribution for the reference data (thought to be highly specific to severe malaria, green triangles, n = 1704, summarised in *Table 1*) and for the Kenyan case data (pink squares, n = 2220; black diamonds: HbAS). The dashed ellipses show the 50% and 95% bivariate normal probability contours approximating each dataset (dark green: reference data; purple: Kenyan data). Panel (**B**) shows the relationship between platelet counts and plasma *Pf*HRP2 in adults with severe malaria from Bangladesh (green circles, n = 172, the dashed green line shows a linear fit) and in children enrolled in the FEAST trial (n = 567, not specific to severe malaria, *Maitland et al., 2011*). Undetectable plasma *Pf*HRP2 concentrations were set to 1 ng/mL ± random jitter. Orange squares: malaria-positive blood slide; black triangles: malaria-negative blood slide. The brown line shows a spline fit to the FEAST data (*smooth.spline* function in R with default parameters) including the data points where *Pf*HRP2 was below the lower limit of detection.

**Table 1.** Summary of severe disease datasets used in our analyses.

For age and parasite density, we show the median values as the distributions are highly skewed. *For the FEAST trial, the severe malaria reference dataset only included platelet and white count data from the 121 patients who had *Pf*HRP2 >1000 ng/mL and >1000 parasites per µL. IQR: interquartile range.

|  | Bangladesh-Thailand | Vietnam | FEAST (Uganda) | Kenya |
|---|---|---|---|---|
| Description | Observational studies of severe malaria | Randomised controlled trials in severe malaria | Randomised controlled trial in severe febrile illness | Observational severe malaria cohort |
| Purpose | Reference data | Reference data | Reference data* and *Figure 1B* | Testing data |
| Published references | *Leopold et al., 2019* | *Hien et al., 1996*; *Phu et al., 2010* | *Maitland et al., 2011* | *MalariaGEN Consortium et al., 2018* |
| $n$ | 653 | 930 | 567 | 2220 |
| Age (years, range) | 28 (2–80) | 30 (15–79) | 2.1 (0–12) | 2.3 (0–13) |
| Parasite density (per µL, IQR) | 48,984 (8289–187,395) | 83,084 (13,047–316,512) | 400 (0–53,200) | 72,000 (6208–315,250) |
| Mortality (%) | 18.2 | 12.9 | 11.3 | 11.6 |

predictive of blood stage parasitaemia and elevated *Pf*HRP2 (p=10^{-16} for a spline term on the $log_{10}$ platelet count in a generalised additive logistic regression model predicting *Pf*HRP2 >1000 ng/mL, Appendix 2). Children enrolled in the FEAST trial who had significant thrombocytopenia (<100,000 platelets per µL) had comparable *Pf*HRP2 concentrations to Asian adults diagnosed with severe falciparum malaria (*Figure 1B*).

## Estimating the proportion of children mis-diagnosed with severe malaria

We can consider the hospitalised Kenyan children in this series as a mixture of two latent sub-populations, 'severe malaria' and 'not severe malaria' (i.e an alternative aetiology for severe illness). To estimate the proportion of each, we use the distribution of HbAS, the human polymorphism most protective against all forms of clinical falciparum malaria. HbAS provides at least 90% protection against severe malaria (*Taylor et al., 2012*; *Malaria Genomic Epidemiology Network, 2014*). The causal SNP rs334 was genotyped in 2213 of the Kenyan children, of whom 57 were HbAS. The causal pathways (a) or (b) in *Figure 2* (note all children have been selected into the study on the basis of clinical symptoms consistent with severe malaria) show how the distribution of HbAS can be used to infer the marginal probability P(Severe malaria) in the Kenyan cohort as the prevalence of HbAS is expected to differ in the two latent sub-populations.

We assumed that cases with the highest likelihood values P(Data | Severe malaria) under the reference model (a bivariate *t*-distribution fit to the severe malaria reference data) had a diagnosis of severe malaria that was 100% specific (top 40% of cases, a sensitivity analysis varied this threshold). The cases with lower likelihood values were assumed to be drawn from a mixture of the two latent populations with an unknown mixing proportion; the prevalence of HbAS in the 'not severe malaria' subgroup was estimated from a cohort of hospitalised children enrolled in the same hospital and who were malaria blood slide positive but were clinically diagnosed as not having severe malaria (n = 6748 of whom 364 were HbAS; *Uyoga et al., 2019*). We assumed that this diagnosis of 'not severe malaria' was 100% specific. Under these assumptions, we estimated that P(Severe malaria) = 0.64 (95% credible interval [C.I.] 0.46–0.8), implying that approximately one-third of the 2200 cases are from the 'not severe malaria' sub-population (they have malaria parasitaemia in addition to another severe illness – likely to be bacterial sepsis – *Figure 2*).

## Estimating individual probabilities of severe malaria

We then estimated P(Severe malaria | Data) for each Kenyan case by fitting a mixture model to the reference data and to the Kenyan data jointly. The model assumed that the platelet and white count data for the Kenyan children were drawn from a mixture of P(Data | Severe malaria) and P(Data | Not severe malaria). The reference data (Asian adults and children with severe malaria and African children with *Pf*HRP2 >1000 ng/mL) were assumed to be drawn only from P(Data | Severe malaria). P

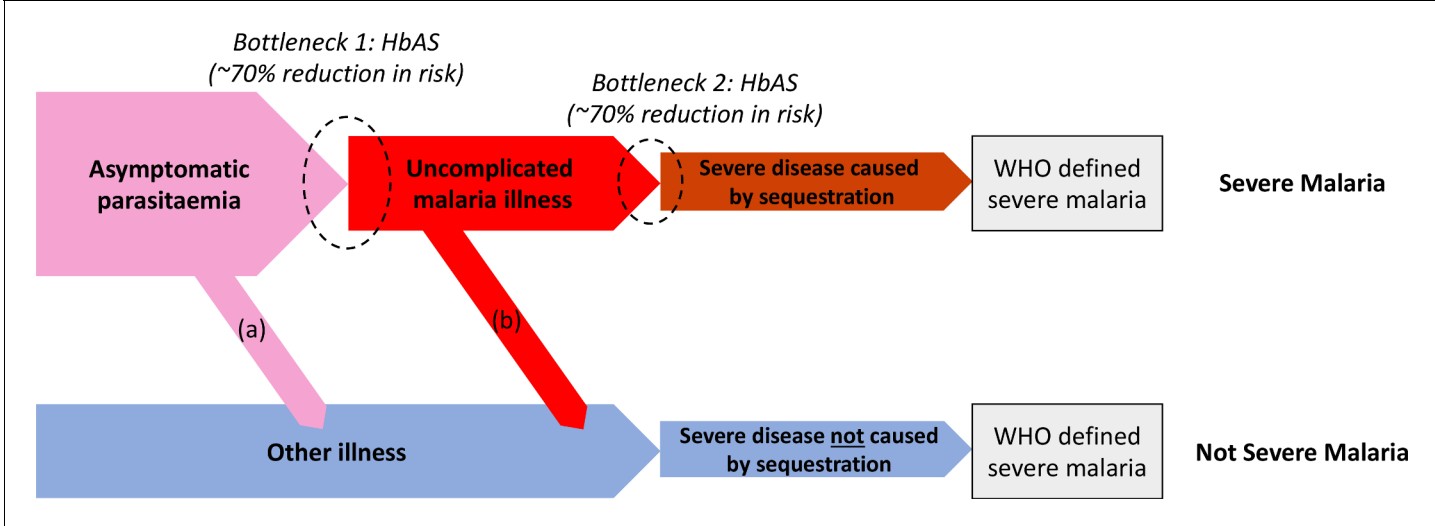

**Figure 2.** Theoretical causal pathways that lead to the clinical diagnosis of severe malaria under the current WHO definition (**World Health Organisation, 2014**). Pathways (a) and (b) represent the two ways patients can be mis-classified as severe malaria. For both pathways (a) and (b), we expect a higher prevalence of HbAS relative to the population with true severe malaria as a consequence of the protective bottlenecks. In this causal model, we assume that HbAS does not protect against asymptomatic parasitaemia, although this assumption is not strictly necessary. Adapted with permission from **Small et al., 2017**.

(Data | Not severe malaria) was modelled itself as a mixture of bivariate *t*-distributions. We used an informative prior on the mixture proportion ('severe malaria' versus 'not severe malaria') in the Kenyan cases, a beta distribution approximating the posterior estimate from the analysis of HbAS prevalence.

*Figure 3A* shows the bimodal distribution of the posterior individual estimates of P(Severe malaria | Data). As expected, the individual posterior probabilities of severe malaria were highly predictive of HbAS ($p = 10^{-6}$ from a generalised additive logistic regression model fit, *Figure 3C*). The individual probabilities were also predictive of in-hospital mortality ($p = 10^{-9}$ from a generalised additive model fit; *Figure 3D*) and admission peripheral blood parasite density ($p = 10^{-25}$ from a generalised additive model fit; *Figure 3E*). In the top quintile of patients with the highest estimated P(Severe malaria | Data), the prevalence of HbAS was 0.7% (3 out of 446). In contrast, for patients in the lowest quintile of estimated P(Severe malaria | Data), the prevalence of HbAS was 4.8% (21 out of 444). The patients with a low probability of severe malaria had a substantially higher case fatality ratio (18.8% mortality for patients in the bottom quintile of P[Severe malaria | Data] versus 6.1% mortality for the top quintile of P[Severe malaria | Data]). This may be explained by the higher case-specific mortality of severe bacterial sepsis (the most likely alternative cause of severe illness). The admission parasite densities in patients with a probability of severe malaria close to 1 were approximately fivefold higher than in patients with a probability of severe malaria close to 0. The blood culture positive rate was 2.1% in the top quintile of P(Severe malaria | Data) and 4.4% in the lowest quintile of P(Severe malaria | Data), and the individual probabilities were predictive of blood culture results ($p = 0.004$ under a generalised additive logistic regression model fit).

## Accounting for case imprecision in case-control studies

'False-positive' cases reduce statistical power and dilute effect size estimates in case-control studies. We propose a novel approach for case-control studies with phenotypic imprecision based on data-tilting (**Nie et al., 2013**). The idea is to 'tilt' the cases towards a pseudo-population with higher specificity for severe malaria. We can do this by re-weighting the data by the probabilities P(Severe malaria | Data), that is, re-weighting the contribution to the log-likelihood in an association model.

We applied this approach as proof of concept to a genome-wide association study using the subset of Kenyan children who had clinical and genome-wide data available (after quality control checks

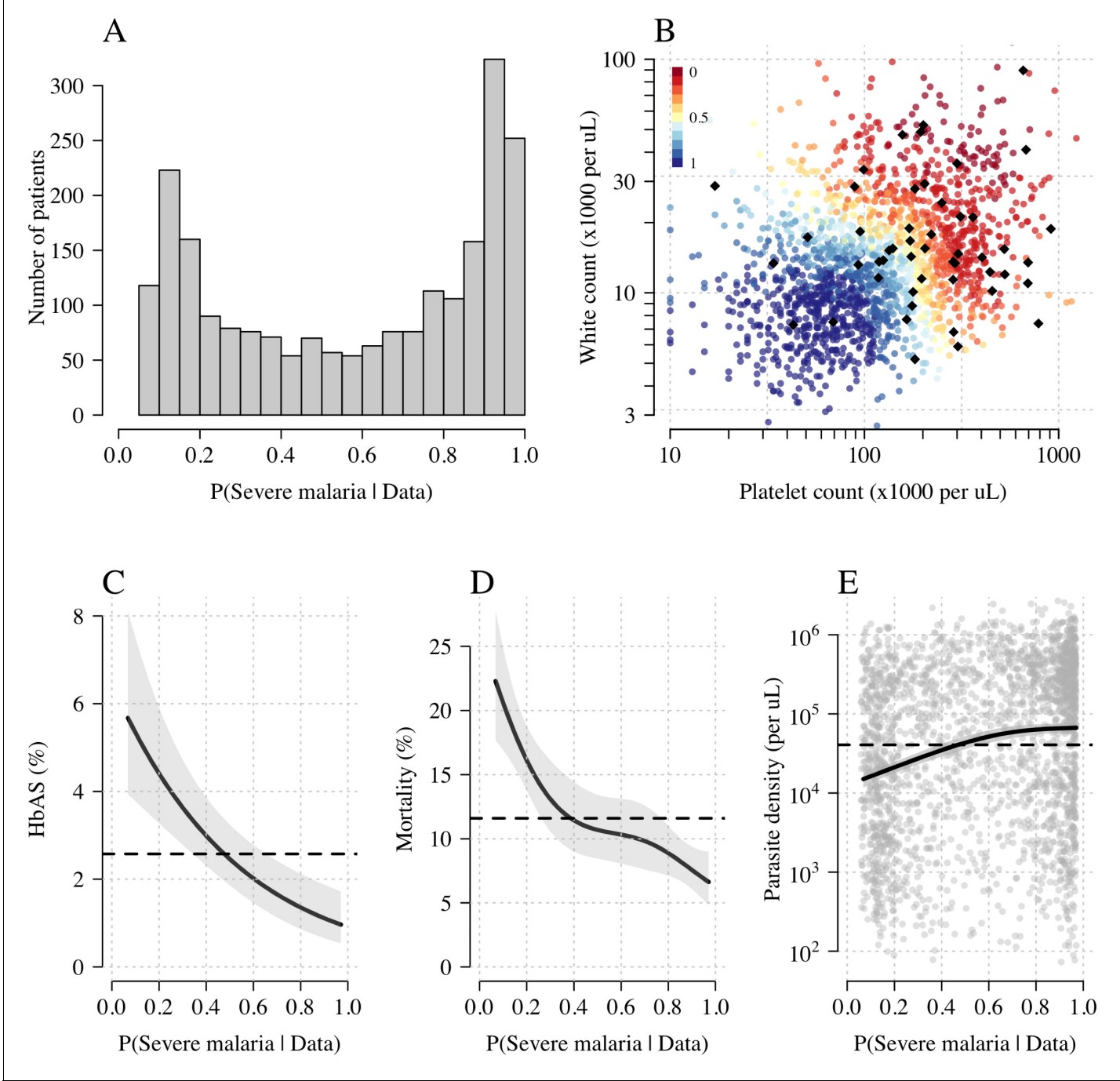

**Figure 3.** Model estimates of P(Severe malaria | Data) in 2220 Kenyan children clinically diagnosed with severe malaria. Panel (**A**) shows the distribution of posterior probabilities of severe malaria being the correct diagnosis. Panel (**B**) shows these same probabilities plotted as a function of the platelet and white counts on which they are based (dark red: probability close to 0; dark blue: probability close to 1). The black diamonds show the HbAS individuals. Panels (**C–E**) show the relationship between the estimated probabilities of severe malaria and HbAS, in-hospital mortality and admission parasite density, respectively. The black lines (shaded areas) show the mean estimated values (95% confidence intervals) from a generalised additive logistic regression model with a smooth spline term for the likelihood (R package *mgcv*). The horizontal lines in panels (**C–E**) show the mean values in the data.

n = 1297 cases) and a set of matched population controls (n = 1614), across 9.6 million biallelic variants on the autosomal chromosomes (*Band et al., 2019*). We compared the data-tilting method to the standard non-weighted approach by estimating local FDRs (*Storey, 2002*). Compared to the standard non-weighted GWAS, data-tilting substantially increased the number of significant associations for local FDRs in the range of 1–5% (*Figure 4*). For example, at an FDR of 2%, the number of significant hits is more than doubled with the additional hits all around known loci associated with protection from severe malaria. We note that if the data weights were not predictive of the true latent phenotype, we would expect fewer significant hits for a given FDR because of the reduction in effective sample size. This is demonstrated by permuting the data weights (for the cases only), which results in 50–75% reduction in the number of significant hits at FDRs < 5% (Appendix 3).

Examining three major genetic regions strongly associated with protection from severe malaria in East Africa (*HBB*: HbAS; *ABO*: O blood group; *FREM3*: in close linkage with the GYPA/B/E structural variants that encode the Dantu blood group; *Band et al., 2019*), the data-tilting approach estimated larger effect sizes compared to the non-weighted model in all three regions (effect size increases: 30% around *HBB*, 9% around *ABO* and 5% around *FREM3*). This resulted in larger $-log_{10}$ p-values for *HBB* and *ABO*, but slightly smaller for *FREM3* (*Figure 5*). We note that there was no signal of association at *ATP2B4* in this subset, most likely due to limited power (*ATP2B4* had the third largest Bayes factor for association in the largest multicentre GWAS to date, *Band et al., 2019*).

## Reappraisal of directly typed polymorphisms

We re-analysed case-control associations for 120 polymorphisms on 70 candidate malaria-protective genes which were typed directly in the 2220 Kenyan children along with 3940 population controls. In this case-control cohort, 14 polymorphisms had previously been identified as associated with protection or increased risk in severe malaria (*MalariaGEN Consortium et al., 2018*). A re-analysis of these 14 variants using the same models of association as previously published and down-weighting the likely mis-classified cases replicated the majority of associations, with increased effect sizes and

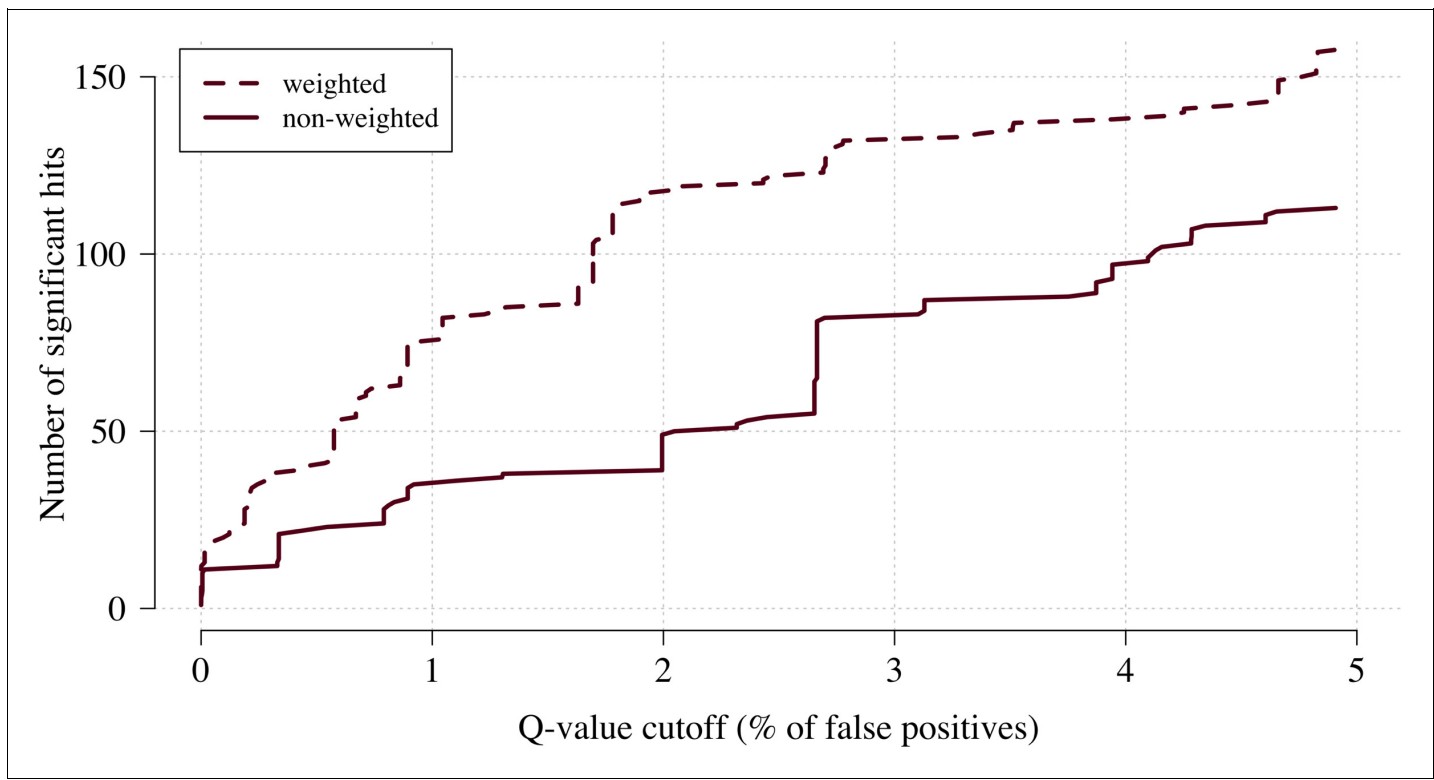

**Figure 4.** The number of significant hits as a function of the FDR for the genome-wide association study across 9.6 million biallelic variants. This analysis is based on a subset of the Kenyan children with whole-genome data available and passing quality checks n = 1297 and n = 1614 controls. Dashed line: weighted model; thick line: non-weighted model.

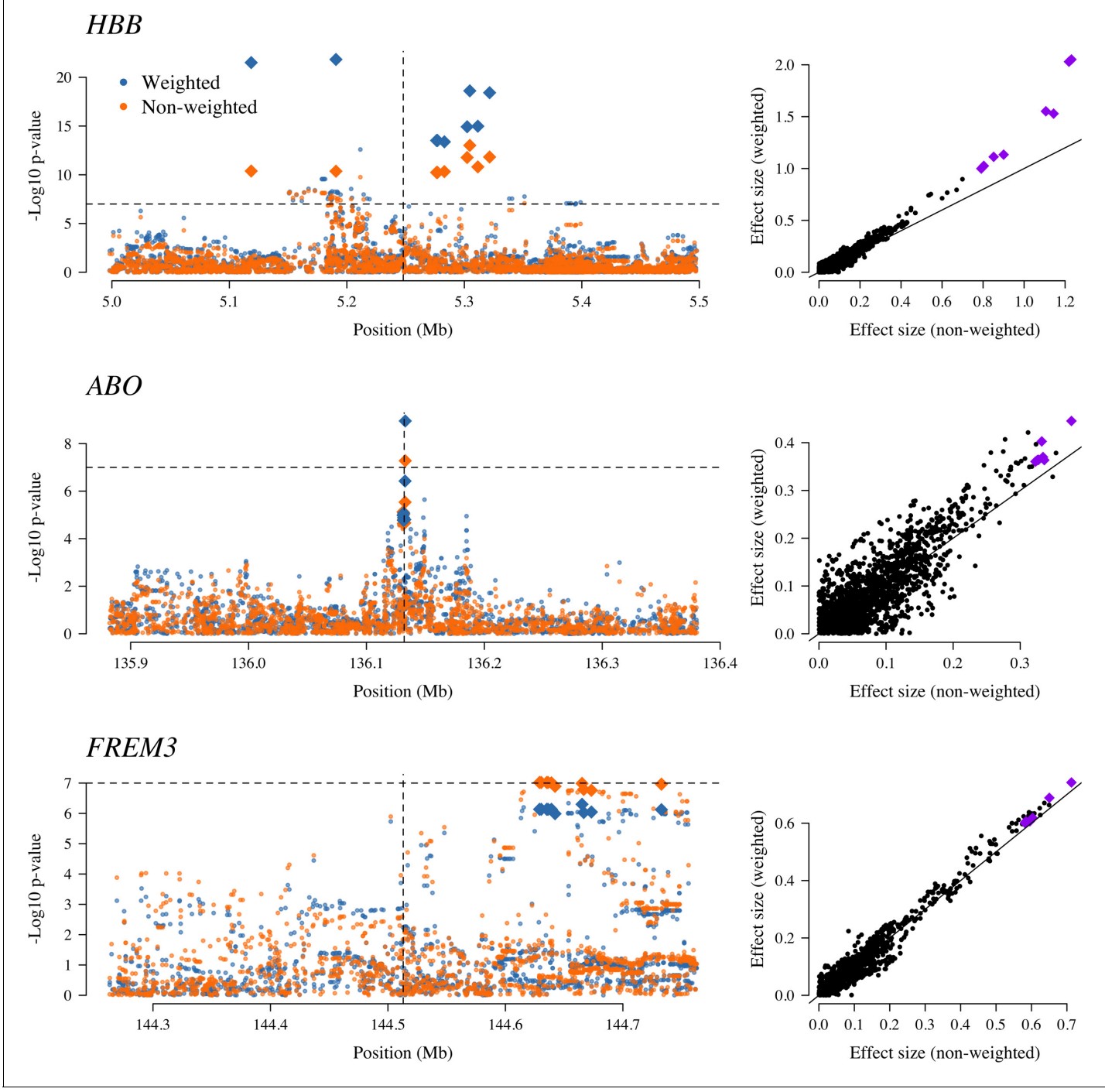

**Figure 5.** The three regions in the human genome with the greatest evidence for protection against severe malaria in East Africa (*HBB*, *ABO* and *FREM3*; *Band et al., 2019*). The Manhattan plots (left panels) compare p-values from the weighted model (blue) and the non-weighted model (orange). Each Manhattan plot is centred around the known causal position shown by the vertical dashed line (0.5 Mb region). The horizontal dashed line shows $p = 10^{-7}$ (threshold often used for defining genome-wide significance). The 10 positions with the greatest $-log_{10}$ p-values under the non-weighted model are shown as large diamonds. The scatter plots on the right compare absolute effect size estimates under both models with the same top 10 hits shown by the larger purple diamonds. Increases of 30, 9 and 5% are seen for the 10 top hits for *HBB*, *ABO* and *FREM3*, respectively.

increased $-log_{10}$ p-values (Appendix 4). For the three major genes (*HBB*, *ABO*, *FREM3*), effect sizes were increased by 10–30% and associations all had higher significance levels on the $-log_{10}$ scale (0.25–1.7). The allele frequencies of all three polymorphisms were directly associated with the probability weights, showing increased protection in individuals more likely to have severe malaria (Appendix 5). Two polymorphisms on the genes *ARL14* and *LOC727982*, reported previously as associated with protection in severe malaria (neither of which are related to red cells), showed decreased effect sizes and $-log_{10}$ p-values and are thus potentially spurious hits.

We explored whether there was evidence of differential effects in the Kenyan cases using P [Severe malaria | Data] to assign probabilistically each case to the 'severe malaria' versus 'not severe malaria' sub-populations. We fitted a categorical logistic regression model predicting the latent sub-population label versus control, where the latent case label was estimated from the weights shown in *Figure 3A*. This resulted in approximately 1279 cases in the 'severe malaria' sub-population and 941 cases in the 'not severe malaria' sub-population. Differential effects were tested by comparing the estimated log-odds for the two sub-populations. After accounting for multiple testing, two polymorphisms showed significant differential effects: rs334 (derived allele encodes haemoglobin S, $p = 10^{-6}$) and rs1050828 (derived allele encodes *G6PD* + 202T, $p = 10^{-3}$ in the model fit to females only), see *Figure 6*. As expected, rs334 was associated with protection in both sub-populations (*Scott et al., 2011*; *Uyoga et al., 2019*) but the effect was almost eight times larger on the log-odds scale in the 'severe malaria' sub-population relative to the 'not severe malaria' sub-population (odds ratio of 0.029 [95% C.I. 0.0088–0.094] in the 'severe malaria' population versus 0.63 [95% C.I. 0.48–0.83] in the 'not severe malaria' population). For rs1050828 (*G6PD* + 202T allele), approximately the same absolute log-odds were estimated for both sub-populations but they had opposite signs. Under an additive model in females, the rs1050828 T allele was associated with protection in the 'severe malaria' sub-population (odds ratio of 0.71 [95% C.I. 0.57–0.88]) but with increased risk in the 'not severe malaria' sub-population (odds ratio of 1.30 [95% C.I. 1.00–1.70]). The additive model including both males and females was consistent with these opposing effects but significant only at a nominal threshold ($p = 0.02$). Opposing effects across the two sub-populations are consistent with the hypothesis that G6PD deficiency leads to a greater risk of being erroneously classified as severe malaria as under the severe anaemia criterion (*Watson et al., 2019*), shown in more detail in Appendix 5. Investigation of haemoglobin concentrations as a function of P(Severe malaria | Data) indicates that the mis-classified group is very heterogeneous, but with a larger proportion of severe anaemia (<5 g/dL) relative to the correctly classified sub-population (Appendix 6).

## Discussion

The clinical diagnosis of severe falciparum malaria in African children is imprecise (*Taylor et al., 2004*; *Bejon et al., 2007*; *White et al., 2013*). Even with quantitation of parasite densities, specificity is still imperfect (*Bejon et al., 2007*). In children with cerebral malaria (unrouseable coma with malaria parasitaemia), the most specific of the severe malaria clinical syndromes, postmortem examination revealed another diagnosis in a quarter of cases studied in Blantyre, Malawi (*Taylor et al., 2004*). Diagnostic specificity can be improved by visualisation of the obstructed microcirculation in vivo (e.g. through indirect ophthalmoscopy) or from parasite biomass indicators (quantitation and staging of malaria parasites on thin blood films, counting of neutrophil-ingested malaria pigment, measurement of plasma concentrations of *Pf*HRP2 or parasite DNA), but these are still largely research procedures and have not been widely adopted or measured at scale for genetic association studies. Our results suggest that imprecision in clinical phenotyping is more substantial than thought previously. In this cohort of 2220 Kenyan children diagnosed with severe malaria from an area of moderate transmission, a probabilistic assessment suggests that around one-third may not have had severe malaria (although malaria may have contributed to their illness; *Small et al., 2017*). This supports our previous conclusion that differences in treatment effects between Asian adults and African children (i.e the benefits of artesunate over quinine in severe malaria estimated from randomised trials; *Dondorp et al., 2005*; *Dondorp et al., 2010*) are predominantly driven by differences in diagnostic specificity (*Hendriksen et al., 2012*; *White et al., 2013*). Mortality was higher in the severe 'not malaria' patients, probably because the main illness was bacterial sepsis. This strongly supports current recommendations to give broad-spectrum antibiotics to all children in endemic areas with suspected severe malaria (*World Health Organisation, 2014*). Using HbAS as a natural experiment

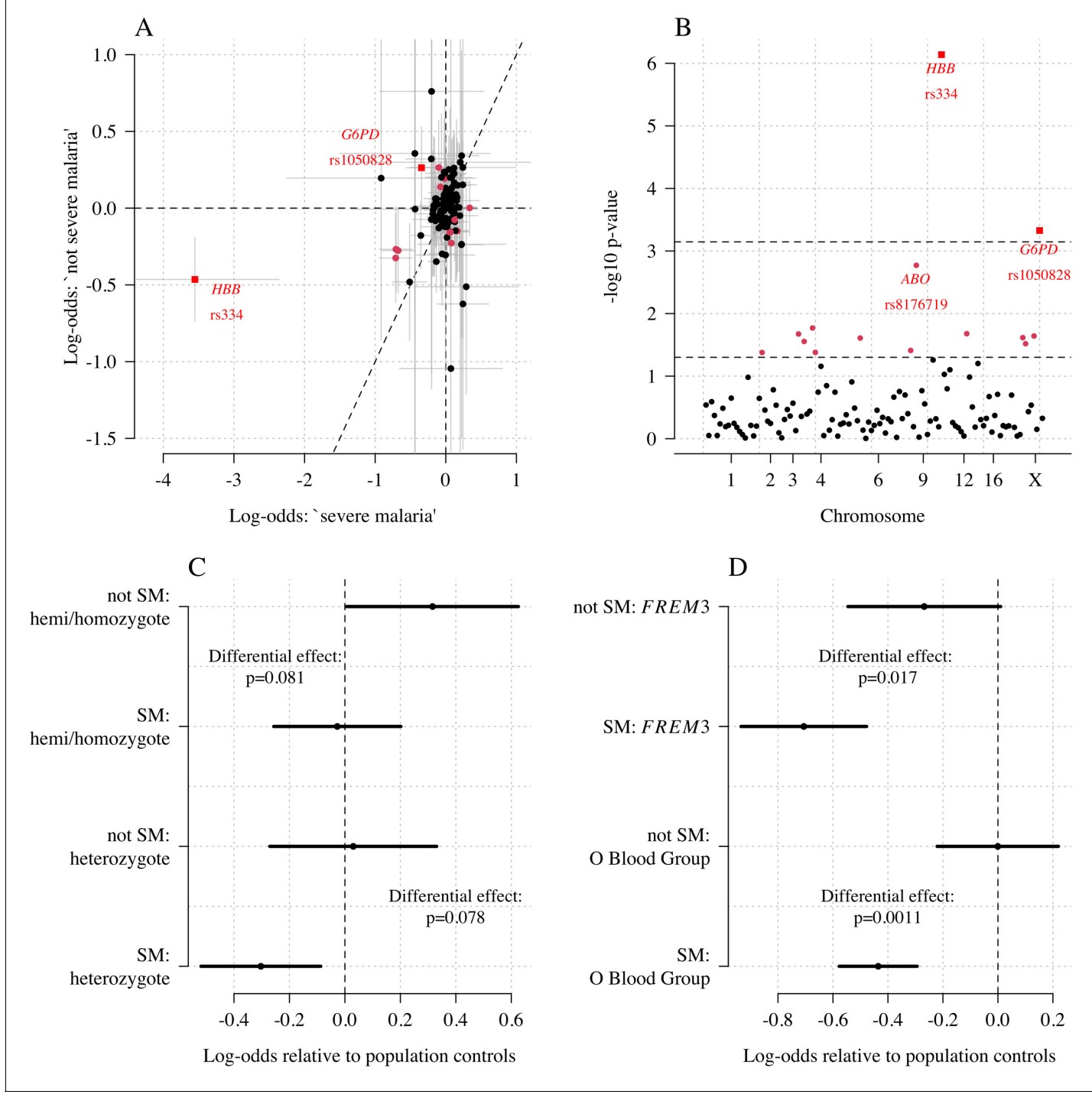

**Figure 6.** Exploring differential effects in 120 directly typed polymorphisms across 70 candidate malaria-protecting genes. (**A**) Case-control effect sizes estimated for the 'severe malaria' sub-population versus the 'not severe malaria' sub-population (n = 3940 controls and n = 2220 cases, with approximately 1279 in the 'severe malaria' sub-population and 941 in the 'not severe malaria' sub-population). The vertical and horizontal grey lines show the 95% credible intervals. (**B**) The $log_{10}$ p-values testing the hypothesis that the effects are the same for the two sub-populations relative to controls. The top dashed line shows the Bonferroni corrected $\alpha = 0.05$ significance threshold (assuming 70 independent tests). The bottom dashed line shows the nominal $\alpha = 0.05$ significance threshold. In both panels, red circles denote $p < 0.05$ (nominal significance level), and red squares denote $p < 0.05/70$. (**C**) Analysis of the rs1050828 SNP (encoding G6PD + 202T) under a non-additive model (hemi/homozygotes and heterozygotes are distinct categories). This shows that heterozygotes are clearly under-represented in the 'severe malaria' sub-population and hemi/homozygotes are clearly over-represented in the 'not severe malaria' sub-population. (**D**) Evidence of differential effects for the O blood group (rs8176719, recessive model) and *FREM3* (additive model).

to validate the biomarker model, we show that the joint distribution of platelet and white blood cell counts is a diagnostic predictor of severe malaria. Complete blood counts are inexpensive and increasingly available in low-resource setting hospitals. Application of an upper threshold of 200,000 platelets per μL would have substantially decreased mis-classification in this large cohort of Kenyan children diagnosed with severe malaria.

This re-analysis using rich clinical data provides additional evidence for the three major genetic polymorphisms protective against severe malaria present in East Africa. After probabilistic down-weighting of the likely mis-classified cases, substantial increases in effect sizes were found. Dilution of effect sizes resulting from mis-classification could partially explain the large heterogeneity in effects noted in the largest severe malaria GWAS to date (*Band et al., 2019*). For haemoglobin S (rs334), there was a fourfold variation in estimated odds ratios across participating sites. Some of this heterogeneity can be attributed to variations in linkage disequilibrium affecting imputation accuracy (*Malaria Genomic Epidemiology Network et al., 2013*), but our analysis shows an additional substantial source of heterogeneity which results from diagnostic imprecision. This can be adjusted for if detailed clinical data are available. For example, in the case of rs334 (directly typed), the data-tilting approach results in a 25% increase in effect size on the log-odds scale, corresponding to 35% decrease in estimated odds ratios (0.1 versus 0.16).

As for the interpretation of genetic effects, one of the most interesting results concerns the *G6PD* gene. G6PD deficiency is the most common enzymopathy of humans. Its potential role in protecting against falciparum malaria has been controversial (*MalariaGEN Consortium et al., 2017*; *Watson et al., 2019*). A very large multi-country genetic association study with over 11,000 severe malaria cases and 17,000 population controls found no overall protective effect of the *G6PD* + 202T allele (the most common mutation in sub-Saharan Africa causing G6PD deficiency), under an additive model (*Malaria Genomic Epidemiology Network, 2014*). The same pattern is observed in this Kenyan cohort (which is a subset of the larger study). In the Kenyan cohort overall, a previous analysis found no clear evidence of protection for male homozygotes but substantial evidence of protection for female heterozygotes (*MalariaGEN Consortium et al., 2015*). This would suggest a heterogyzote advantage leading to a balancing polymorphism. However, when the Kenyan cases are modelled as two distinct sub-populations, there is evidence of differential effects between the 'severe malaria' and 'not severe malaria' sub-populations. Hemi- and homozygous G6PD deficiency was associated with an increased risk of mis-classification (reflecting an increased risk of severe anaemia), but it is unclear whether or not hemi/homozygous G6PD deficiency was protective in the 'true severe malaria' sub-population (*Figure 6C*). On the other hand, heterozygote deficiency was very clearly protective in the true severe malaria subgroup, consistent with previous findings, and did not appear to lead to an increased risk of mis-classification (consistent with a lower risk of extensive haemolysis and thus false classification in heterozygotes who have both normal and G6PD-deficient erythrocytes in their circulation). When examining the 'severe malaria' sub-population only, the sample size in this study is too small to discriminate between the heterozygote and additive models of association. In our view, the relationship between G6PD deficiency and severe falciparum malaria remains unanswered. A biomarker-driven approach should be applied to other case-control cohorts for a definitive understanding of the role of this major human polymorphism.

The limitations of our diagnostic model can be summarised as follows. First, the validity and interpretation of the individual probabilities of severe malaria is heavily dependent on the reference model and thus the reference data. Our reference data were primarily from Asian adults in whom diagnostic specificity for severe malaria is thought to be very high. Diagnostic checks suggested that the marginal distributions of platelet counts were similar between adults and children, and we made age corrections to the white blood cell count, but small deviations could reduce the discriminatory value (e.g. lower white counts associated with the Duffy negative phenotype; *Reich et al., 2009*). Second, it is possible that rare genetic conditions exist in which the probabilities of severe malaria under this model might be biased. One example is sickle cell disease (HbSS, <0.5% in the Kenyan cases), which results in chronic inflammation with high white counts and low platelet counts relative to the normal population (*Sadarangani et al., 2009*). The 11 children with HbSS in this cohort were all assigned low probabilities of severe malaria, but this should be interpreted with caution. Whether HbSS is protective against severe malaria or increases the risk of severe malaria remains unclear (*Williams and Obaro, 2011*). For these patients, other biomarkers such as plasma *Pf*HRP2 may be more appropriate. Third, it is possible that the joint distribution of the complete blood count

variables used to fit the reference model could be dependent on the severe malaria sub-phenotype. For example, if the reference data were biased towards cerebral malaria, and the joint distribution of platelet and white cell counts in cerebral malaria differed from those in the other severe malaria syndromes, then the predicted outliers could represent other forms of severe malaria instead of 'not severe' malaria. However, there are no known biological reasons why this would be the case. The strong correlation between platelet counts and *Pf*HRP2 (*Figure 1B*) suggests that low platelet counts are a universal feature of severe malaria.

In summary, under a probabilistic model based on routine blood count data, we have shown that it is possible to estimate mis-classification rates in diagnosed severe childhood malaria in a malaria endemic area of East Africa and compute probabilistic weights that can downweight the contribution of likely mis-classified cases. The well-established protective effect of HbAS provided an independent validation of the model. Relative to predicted mis-classified cases, patients predicted to have 'true severe malaria' had a substantially lower prevalence of HbAS, higher parasite densities, lower rates of positive blood cultures and lower mortality. These data strongly support the current guideline to give broad-spectrum antibiotics to all children with suspected severe malaria and suggest that normal range platelet counts (>200,000 per μL) could be used as a simple exclusion criterion in studies of severe malaria. Based on this analysis, we recommend that future studies in severe malaria collect and record complete blood count data. Further studies of platelet and white blood cell counts from a diverse cohort of children with severe falciparum malaria, confirmed using high-specificity diagnostic techniques such as visualisation of the microcirculation, and measurement of plasma *Pf*HRP2, or plasma *P. falciparum* DNA concentrations, should be conducted to validate this approach.

## Materials and methods

### Data

#### Kenyan case-control cohort

The Kenyan case-control cohort has been described in detail previously (*MalariaGEN Consortium et al., 2018*). Severe malaria cases consisted of all children aged <14 years who were admitted with clinical features of severe falciparum malaria to the high-dependency ward of Kilifi County Hospital between 11 June 1999 and 12 June 2008. Severe malaria was defined as a positive blood film for *P. falciparum* along with prostration (Blantyre Coma Score of 3 or 4), cerebral malaria (Blantyre Coma Score of <3), respiratory distress (abnormally deep breathing) and severe anaemia (haemoglobin <5 g/dL). Controls were infants aged 3–12 months who were born within the same area as the cases and who were recruited to a cohort study investigating genetic susceptibility to a wide range of childhood diseases. Cases and controls were genotyped for the rs334 SNP and for $\alpha^+$-thalassaemia along with 120 other SNPs using DNA extracted from fresh or frozen samples of whole blood as described in detail previously (*MalariaGEN Consortium et al., 2018*; *Wambua et al., 2006*).

#### Fluid Expansion as Supportive Therapy (FEAST)

FEAST was a multicentre randomised controlled trial comparing fluid boluses for severely ill children (n = 3161) that was not specific to severe malaria (*Maitland et al., 2011*). Platelet counts, white blood cell counts, parasite densities and *Pf*HRP2 were jointly measured for 566 children (patients enrolled in the sites in Mulago, Lacor and Mbale, in Uganda). In order to select only those with a very high probability of having severe malaria as the primary cause of illness, we selected the 121 children who had measured *Pf*HRP2 >1000 ng/mL and parasitaemia >1000 per μL.

#### AQ Vietnam and AAV randomised controlled trials

The AQ and the AAV studies were two randomised clinical trials in Vietnamese adults diagnosed clinically with severe falciparum malaria recruited to a specialist ward of the Hospital for Tropical Diseases, Ho Chi Minh City, Vietnam, between 1991 and 2003 (*Hien et al., 1996*; *Phu et al., 2010*). AQ Vietnam was a double-blind comparison of intramuscular artemether versus intramuscular quinine (n = 560); AAV compared intramuscular artesunate and intramuscular artemether (n = 370).

## Observational studies in Thai and Bangladeshi adults and children

We included data from multiple observational studies in severe falciparum malaria conducted by the Mahidol Oxford Tropical Medicine Research Unit in Thailand and Bangladesh between 1980 and 2019. These pooled data have been described previously (*Leopold et al., 2019*). Platelet counts and white blood cell counts were available in 657 patients. We excluded one 30-year-old adult from Bangladesh whose recorded platelet count was 1000 per µL and three other adults with platelet counts greater than 450,000 per µL as outliers reflecting likely data entry errors. Plasma *Pf*HRP2 concentrations were available in 172 patients from Bangladesh. 55 patients from this series were younger than 15 years of age.

## Multiple imputation

In the Kenyan severe malaria cohort (n = 2220), data on platelet counts were missing in 18%, white blood counts were missing in 0.2% and parasite density was missing in 1.6%. In-hospital outcome (died/survived) was missing for 13 patients. rs334 genotype was missing for 7; $\alpha^+$-thalassaemia genotype was missing for 101 patients. In the Vietnamese adults, platelet counts were missing in 4%, white counts in 2% and parasitaemia in 0%.

We did multiple imputation using random forests for all available clinical variables using the R package *missForest* (targeted genotyping data was not included for imputation). Appendix 7 shows the missing data pattern in the studies in Vietnamese adults and in the Kenyan severe malaria cases. Ten datasets were imputed for each dataset independently and were used for the subsequent analyses. Analyses using directly typed genetic polymorphisms or the within-hospital outcome as the dependent variables used only the data where these outcomes were recorded, assuming that they were missing at random.

## Reference model of severe malaria

### Biological rationale

Thrombocytopenia accompanied by a normal white blood count and a normal neutrophil count are typical features of severe malaria (*Hanson et al., 2015*; *Leblanc et al., 2020*), but they may also occur in some systemic viral infections and in severe sepsis. Neutrophil leukocytosis may sometimes occur in very severe malaria, but is more characteristic of pyogenic bacterial infections. These indices, whilst individually not very specific, could each have useful discriminatory value. We reasoned therefore that their joint distribution could help discriminate between children with severe malaria versus those severely ill with coincidental parasitaemia. The Kenyan severe malaria cohort did not have differential white count data, so we used platelet counts and total white blood cell counts as the two diagnostic biomarkers in the reference model of severe malaria.

### Choice of reference data and confounders

The best data for fitting the biomarker model are either from children or adults from low transmission areas (where parasitaemia has a high positive predictive value) or in children or adults with high plasma *Pf*HRP2 measurements indicating a large latent parasite biomass (*Hendriksen et al., 2012*).

In the first years of life, white blood cell counts are often much higher than in adults because of lymphocytosis. We used data from 858 children from the FEAST trial, in whom white counts were measured, to estimate the relationship between age and mean white count in severe illness (median age was 24 months). The estimated relationship is shown in Appendix 8 (using a generalised additive linear model with the white count on the $log_{10}$ scale), with mean white counts reaching a plateau around 5 years of age. We used this to correct all white count data in children less than 5 years of age, both in the reference data and the Kenyan cohort.

There is also a systematic difference associated with the Duffy negative phenotype which is near fixation in Africa but absent in Asia. Duffy negative individuals have lower neutrophil counts (termed benign ethnic neutropenia) (*Reich et al., 2009*). The use of Asian adults to estimate the reference distribution of white counts in severe malaria could thus falsely include individuals with elevated white counts (relative to the normal ranges). However, a diagnostic quantile-quantile plot (Appendix 1, on the log scale) comparing the white blood cell count distribution in Vietnamese adults and in children in the FEAST trial who had *Pf*HRP2 >1000 ng/mL did not suggest any major differences. In fact the African children had slightly higher white counts on average even after the correction for

age. This may represent imperfect specificity for severe malaria when using a plasma *Pf*HRP2 cutoff of 1000 mg/mL.

For platelet counts (which have the greatest diagnostic value for severe malaria in our series), age is not a confounder and published data support the hypothesis that thrombocytopenia is highly specific for 'true' severe malaria in children as well as adults suspected of having severe malaria (with a diagnostic and a prognostic value). The French national guidelines specifically mention thrombocytopenia (<150,000 per µL) for the diagnosis of severe malaria in children who have travelled to a malaria endemic area. In a French paediatric severe malaria series in travellers, almost half had severe thrombocytopenia (<50,000 per µL) (*Lanneaux et al., 2016*; *Pediatric Imported Malaria Study Group for the 'Centre National de Référence du Paludisme' et al., 2017*). In Dakar, Senegal (one of the lowest transmission areas in Africa), thrombocytopenia was an independent predictor of death and the median platelet count was 100,000 (*Gérardin et al., 2007*; *Gérardin et al., 2002*). Comparison of the distributions of platelet counts (on the log scale) between Asian children and Asian adults suggested no major differences (Appendix 1), although we had few data for Asian children. In the seminal Blantyre autopsy study (*Taylor et al., 2004*), platelet counts were substantially different between fatal cases confirmed postmortem to be severe malaria (62,000 per µL and 56,000 per µL for the children with sequestration only and sequestration + microvascular pathology, respectively) and fatal cases with a mis-diagnosis of severe malaria (no sequestration: 176,000 per µL; the inter-group difference was statistically significant, $p = 0.008$). A larger cohort from the same centre in Malawi reported substantially higher platelet counts in retinopathy-negative cerebral malaria (mean platelet count was 161,000 per µL, n = 288) compared to retinopathy-positive cerebral malaria (mean count was 81,000 per µL, n = 438) (*Small et al., 2017*).

We visually checked approximate normality for each marginal distribution using quantile-quantile plots (Appendix 9). On the $log_{10}$ scale, platelet counts and white counts show a good fit to the normal approximation but with some outliers so a *t*-distribution was used (robust to outliers). For all modelling of the joint distribution of platelet counts and white blood cell counts, we chose bivariate *t*-distributions with 7 degrees of freedom as the default model. The final reference model used was a bivariate *t*-distribution fit to the joint distribution of platelet counts and white counts both on the logarithmic scale. On the $log_{10}$ scale, the mean values (standard deviations) were approximately 1.76 (0.11) and 0.92 (0.055) for platelets and white counts, respectively. The covariance was approximately 0.0035. These values varied very slightly across the 10 imputed datasets. Log-likelihood values for each severe malaria case in the Kenyan cohort were calculated for each imputed dataset independently. The median log-likelihoods per case were then used in downstream analyses.

## Limitations of the model

The diagnostic model of severe malaria using platelet counts and white blood cell counts cannot be applied to all patients. We summarise here the known and possible limitations. When using this model to estimate the association between a genetic polymorphism and the risk of severe malaria, if the genetic polymorphism of interest affects the complete blood count independently, there will be selection bias (see the directed acyclic graph in Appendix 10). One example is HbSS. Children with HbSS have chronic inflammation with white blood cells counts about 2–3 times higher than normal and slightly lower platelet counts (*Sadarangani et al., 2009*). All 11 children in the Kenyan cohort with HbSS were assigned low probabilities of having severe malaria (Appendix 10), but these probabilities could reflect a deficiency of the model. Including or excluding these children from the analysis had no impact on the results as they represent less than 0.5% of the cases.

The second possible limitation concerns the validation using HbAS. Previous studies have suggested negative epistasis between the malaria-protective effects of HbAS and $\alpha^+$-thalassaemia (*Williams et al., 2005*; *Opi et al., 2014*). The 3.7 kb deletion across the *HBA1-HBA2* genes (known as $\alpha^+$-thalassaemia) has an allele frequency of $\sim 40\%$ in this population; therefore, 16% of HbAS individuals are homozygous for $\alpha^+$-thalassaemia (*Ndila et al., 2020*). Negative epistasis implies that those with both polymorphisms would have less or no protective effect against severe malaria. Of the 2113 Kenyan cases with both HbS and $\alpha^+$-thalassaemia genotyped, 13 were HbAS and homozygous $\alpha^+$-thalassaemia. Appendix 11 shows that the majority of those with both polymorphisms had clinical indices pointing away from severe malaria, suggesting that the observed number of patients with both HbAS and homozygous $\alpha^+$-thalassaemia is inflated by two- to threefold.

The third possible problem concerns the use of white blood cell counts in relation to invasive bacterial infections. Bacteraemia could either be the cause of severe illness (with coincidental parasitaemia) or it could be concomitant (which may result from extensive parasitised erythrocyte sequestration in the gut), that is, a result of severe malaria. The former should be identified as 'not severe malaria' (as bacteraemia is the main cause of illness), but the latter should be identified as 'severe malaria' and might be mis-classified as 'not severe malaria' under our model. However, in a series of 845 Vietnamese adults (high diagnostic specificity), only one of eight patients who had concomitant-invasive bacterial infections and a white count measured had leukocytosis (median white count was 8100; range 3500–14,850 per µL; *Phu et al., 2020*).

## Estimating the diagnostic specificity in the Kenyan cohort

We assume that the Kenyan cases are a latent mixture of two sub-populations: $P_0$ is the population 'severe malaria' and $P_1$ is the population 'not severe malaria' (mis-classified). For a set of diagnostic biomarkers $X$, this implies that $X \sim G = \pi f_0 + (1 - \pi)f_1$, where $f_0, f_1$ are the sampling distributions (likelihoods) of each sub-population, respectively.

We can infer the value of $\pi$ (proportion correctly classified as severe malaria) without making parametric assumptions about $f_1$ by using the distribution of HbAS (motivated by the causal pathways shown in *Figure 2*). This is done as follows: we first estimate $\widehat{f_0}$ by fitting a bivariate $t$-distribution to the reference data – this approximates the sampling distribution for $P_0$. We then make three assumptions:

1. Out of the 2213 Kenyan cases with rs334 genotyped, we assume that cases in the top 40th percentile of the likelihood distribution under $\widehat{f_0}$ are drawn from $P_0$: $N_0 = 887$, of which $N_0^{sickle} = 9$ are HbAS.
2. For the other cases, the proportion drawn from $P_0$ is unknown and denoted $\pi'$: $N_G = 1,326$, of which $N_G^{sickle} = 48$ are HbAS.
3. Finally, additional information is incorporated by using data from a cohort of individuals with severe disease from the same hospital who had positive malaria blood slides but whose diagnosis was not severe malaria ($N_1 = 6,748$, of which $N_1^{sickle} = 364$ were HbAS) (*Uyoga et al., 2019*).

Under these assumptions, we can fit a Bayesian binomial mixture model to these data with three parameters: $\{\pi', p_0, p_1\}$. The likelihood is given by

$$N_0^{sickle} \sim \text{Binomial}(p_0, N_0)$$
$$N_G^{sickle} \sim \text{Binomial}(\pi'p_0 + (1 - \pi')p_1, N_G)$$
$$N_1^{sickle} \sim \text{Binomial}(p_1, N_1)$$

The priors used were $p_1 \sim \text{Beta}(5, 95)$ (i.e. 5% prior probability with 100 pseudo observations); $p_0 \sim \text{Beta}(1, 99)$ (1% prior probability with 100 pseudo observations). A sensitivity analysis with flat beta priors (Beta[1,1]) did not qualitatively change the result (by one percentage point for the final estimate of $\pi$). To check the validity of the use of the external population from *Uyoga et al., 2019*, we did a sensitivity analysis using the lowest quintile of the likelihood ratio distribution as a population drawn entirely from $P_1$ (instead of the external data from *Uyoga et al., 2019*).

## Estimating P(Severe malaria | Data) in the Kenyan cohort

Denote the platelet and white count data from the FEAST trial as $\{X_i^{\text{FEAST}}\}_{i=1}^{121}$; the data from the Vietnamese adults and children as $\{X_i^{\text{Asia}}\}_{i=1}^{1583}$; the data from the Kenyan children as $\{X_i^{\text{Kenya}}\}_{i=1}^{2220}$. We fit the following joint model to the reference biomarker data and the Kenyan biomarker data.

$$X_i^{\text{FEAST}} \sim \text{Student}(\mu_{SM}^1, \Sigma_{SM}^1, 7)$$
$$X_i^{\text{Asia}} \sim \text{Student}(\mu_{SM}^2, \Sigma_{SM}^2, 7)$$
$$X_i^{\text{Kenya}} \sim \pi f_0 + (1 - \pi)f_1$$
$$f_0 = p\,\text{Student}(\mu_{SM}^1, \Sigma_{SM}^1, 7) + (1 - p)\text{Student}(\mu_{SM}^2, \Sigma_{SM}^2, 7)$$
$$f_1 = \sum_{j=1}^{K} \alpha_j\,\text{Student}(\mu_{notSM}^j, \Sigma_{notSM}^j, 7)$$

with the following prior distributions and hyperparameters, where $\alpha = \{\alpha_1, .., \alpha_K\}$ such that $\sum_{j=1}^{K} \alpha_j = 1$:

$$\pi \sim \text{Beta}(40.3, 24.7)$$
$$p \sim \text{Beta}(2, 2)$$
$$\mu_{SM}^{1,2} \sim \text{Normal}(\{1.8, 0.95\}, 0.1^2)$$
$$\mu_{notSM}^{1..K} \sim \text{Normal}(\{2.5, 1.5\}, 0.25^2)$$
$$\alpha \sim \text{Dirichlet}(1/K, ..., 1/K)$$

The covariance matrices $\Sigma_{SM}^{1,2}$ and $\Sigma_{SM}^{1,6}$ were parameterised as their Cholesky LKJ decomposition, where the L correlation matrices had a uniform prior (i.e. hyperparameter $\nu = 1$). The model was implemented in *rstan*.

This models the biomarker data in 'not severe malaria' as a mixture of $K$ *t*-distributions. We chose $K = 6$ as the default choice (sensitivity analysis increasing this has no impact). The Dirichlet prior with hyperparameter $1/K$ forces sparsity in this mixture model (most of the prior weight is on the vertices of the K-dimensional simplex); see, for example, *Frühwirth-Schnatter and Malsiner-Walli, 2019*. This is a very general and flexible way of modelling the 'not severe malaria' distribution: we are not trying to make inferences about this distribution, we just want the mixture model to be flexible enough to describe it. The model also allows for differences in the joint distribution of platelet counts and white counts between the reference datasets (FEAST trial and the Asian studies). The Kenyan cases drawn from the 'severe malaria' sub-population are then modelled as a mix of these two reference models.

## Re-weighted likelihood for case-control analyses

For each $\{X_i^{\text{Kenya}}\}_{i=1}^{2220}$, we estimate the posterior probability of being drawn from the sampling distribution $f_0$. The mean posterior probability then defines a precision weight $w_i$ which can be used in a standard generalised linear model (glm) with the same interpretation as inverse probability weights. The weighted glm is equivalent to computing the maximum likelihood estimate where the log-likelihood is weighted by $w_i$. In our case-control analyses, all the controls are given weight 1. *Nie et al., 2013* give a proof of correctness for this re-weighted log-likelihood (equivalent to 'tilting' the dataset towards the desired distribution $\widehat{f_0}(X)$). The log-odds ratio computed from the weighted logistic regression can be interpreted as the causal effect of the polymorphism on 'true severe malaria' relative to the controls, where 'true severe malaria' is defined by the sampling distribution $f_0$. Appendix 12 shows the results of a simulation study demonstrating how the effect estimates and standard error estimates vary as a function of the proportion of mis-classified cases (as given by the probability weights).

## Genome-wide association study

Anonymised whole-genome data from the Illumina Omni 2.5M platform for 1944 severe malaria cases and 1738 population controls were downloaded from the European Genome-Phenome Archive (dataset accession ID: EGAD00010001742, release date March 2019; *Band et al., 2019*). This contained sequencing data on 2,383,648 variants. We used the quality control metadata provided with the 2019 data release to select SNPs and individuals with high-quality data. We first excluded 386 individuals (due to relatedness: 155; missing data or low intensity: 226; gender: 5). We then removed 616,426 SNPs that did not pass quality control, leaving a total of 1,767,222 SNPs. We used *plink2* to prune the SNPs (options: –maf 0.01 –indep-pairwise 50 2 0.2) down to a set of 462,120 SNPs in approximate linkage equilibrium. These SNPs were then used to calculated the first five principal components (Appendix 13), which we subsequently used to control for population structure in the genome-wide association study. We used the Michigan imputation server with the 1000 Genomes Phase 3 (version 5) as the reference panel to impute 28.6 million polymorphisms across the 22 autosomal chromosomes. This is a web-based service that runs imputation pipelines (phasing is done with Eagle2, imputation with Minimac4). Encrypted results are returned with a one-time password. Of the remaining 3682 individuals (1681 cases and 1615 controls), we had clinical data available for 1297 cases. We only used the subset of individuals with clinical data available in order for a fair comparison between the weighted and non-weighted genome-wide association studies. We ran subsequent genome-wide association studies on all biallelic sites with a minor allele frequency $\geq 5\%$ (9,615,446 sites in total) assuming an additive model of association. We used the R function *glm* with a binomial link for all tests of association (genetic data were encoded as the

number of ancestral alleles). The supplementary appendix gives the R code for weighted logistic regression. The point estimates from the weighted model estimated by *glm* are correct but it is necessary to transform the standard errors in order to take into account the reduction in effective sample size (see code).

## Case-control study in directly typed polymorphisms

We fit a categorical (multinomial) logistic regression model to the case-control status as a function of the directly typed polymorphisms (120 after discarding those that are monomorphic in this population; see *MalariaGEN Consortium et al., 2018* for additional details). We modelled the severe malaria cases as two separate sub-populations with a latent variable: 'severe malaria' versus 'not severe malaria', resulting in three possible labels (controls, 'severe malaria', 'not severe malaria'). The models adjusted for self-reported ethnicity and sex. The model was coded in *stan* (*Stan Development Team, 2020*) using the log-sum-exp trick to marginalise out the likelihood over the latent variables (see code). Normal(0,5) priors were set on all parameters, and parameter estimates and standard errors were estimated from the maximum a posteriori value (function *optimizing* in *rstan*).

## Code availability

Code, along with a minimal clinical dataset for reproducibility of the diagnostic phenotyping model, is available via a GitHub repository: https://github.com/jwatowatson/Kenyan_phenotypic_accuracy (*Watson, 2021*; copy archived at swh:1:rev:03a2de285d38b85a769aa25de46b7960487efc62).

## Data availability

A curated minimal clinical dataset is currently available alongside the code on the GitHub repository. This will also be made available at publication via the KEMRI-Wellcome Harvard Dataverse (https://dataverse.harvard.edu/dataverse/kwtrp).

This paper used genome-wide genotyping data generated by *Band et al., 2019*, available on request from the European Genome-Phenome Archive (dataset accession ID: EGAD00010001742).

Requests for access to appropriately anonymised clinical data and directly typed genetic variants (*Malaria Genomic Epidemiology Network, 2014*) for the Kenyan severe malaria cohort can be made by application to the data access committee at the KEMRI-Wellcome Trust Research Programme by email to mmunene@kemri-wellcome.org.

The FEAST trial datasets are available from the principal investigator on reasonable request (k.maitland@imperial.ac.uk). Requests for access to appropriately anonymised clinical data from the AQ and AAV Vietnam study and the Asian paediatric cohort can be made via the Mahidol Oxford Tropical Medicine Research Unit data access committee by emailing the corresponding author JAW (jwatowatson@gmail.com) or Rita Chanviriyavuth (rita@tropmedres.ac).

## Acknowledgements

This research was funded by The Wellcome Trust. A CC BY or equivalent licence is applied to the author accepted manuscript arising from this submission, in accordance with the grant's open access conditions. This work was done as part of SMAART (Severe Malaria Africa – A consortium for Research and Trials) funded by a Wellcome Collaborative Award in Science grant (209265/Z/17/Z) held in part by KM, NPJD and AD. TNW and NJW are senior and principal research fellows respectively funded by the Wellcome Trust (202800/Z/16/Z and 093956/Z/10/C, respectively). ECG acknowledges funding from a core grant to the MRC CTU at UCL from the MRC (MC_UU_12023/26).

The human data used in this study was generated through the Malaria Genomic Epidemiology Network (https://www.MalariaGEN.net) Consortial Project 1, for which a full list of Consortium members is provided at https://www.malariagen.net/projects/consortial-project-1/malariagen-consortium-members. The Malaria Genomic Epidemiology Network study of severe malaria was supported by Wellcome (WT077383/Z/05/Z) and the Bill and Melinda Gates Foundation (https://www.gatesfoundation.org/) through the Foundations of the National Institutes of Health (https://fnih.org/) as part of the Grand Challenges in Global Health Initiative. The Resource Centre for Genomic Epidemiology of Malaria is supported by Wellcome (090770/Z/09/Z; 204911/Z/16/Z). This research was

supported by the Medical Research Council (G0600718; G0600230; MR/M006212/1). Wellcome also provides core awards to the Wellcome Centre for Human Genetics (203141/Z/16/Z) and the Wellcome Sanger Institute (206194).

This study also makes use of data from the FEAST trial. The FEAST trial was supported by a grant (G0801439) from the Medical Research Council, UK, provided through the (MRC) DFID concordat. KM and ECG were supported by this grant.

## Additional information

### Funding

| Funder | Grant reference number | Author |
|---|---|---|
| Wellcome Trust | 209265/Z/17/Z | Kathryn Maitland<br>Nicholas PJ Day<br>Arjen M Dondorp |
| Wellcome Trust | 202800/Z/16/Z | Thomas N Williams |
| Wellcome Trust | 093956/Z/10/C | Nicholas J White |
| Medical Research Council | MC\UU\12023/26 | Elizabeth C George |
| Wellcome Trust | WT077383/Z/05/Z | Kirk Rockett |
| Medical Research Council | G0801439 | Elizabeth C George<br>Kathryn Maitland |
| Wellcome Trust | 090770/Z/09/Z 204911/Z/16/Z | Kathryn Maitland |
| Medical Research Council | G0600718 G0600230 MR/M006212/1 | Kathryn Maitland |
| Wellcome Trust | 203141/Z/16/Z | Kathryn Maitland |
| Wellcome Trust | 206194 | Kathryn Maitland |

The funders had no role in study design, data collection and interpretation, or the decision to submit the work for publication.

### Author contributions

James A Watson, Conceptualization, Software, Formal analysis, Investigation, Visualization, Methodology, Writing - original draft, Writing - review and editing; Carolyne M Ndila, Sophie Uyoga, Kirk Rockett, Resources, Data curation, Writing - review and editing; Alexander Macharia, Caroline Ngetsa, Neema Mturi, Norbert Peshu, Stije Leopold, Hugh Kingston, Elizabeth C George, Data curation, Writing - review and editing; Gideon Nyutu, Data curation, Methodology; Shebe Mohammed, Benjamin Tsofa, Data curation; Kathryn Maitland, Data curation, Funding acquisition, Writing - review and editing; Nicholas PJ Day, Philip Bejon, Resources, Supervision, Funding acquisition, Validation, Writing - review and editing; Arjen M Dondorp, Resources, Validation, Writing - review and editing; Thomas N Williams, Resources, Data curation, Supervision, Funding acquisition, Validation, Methodology, Writing - review and editing; Chris C Holmes, Resources, Supervision, Validation, Methodology, Writing - review and editing; Nicholas J White, Conceptualization, Resources, Supervision, Funding acquisition, Validation, Methodology, Project administration, Writing - review and editing

### Author ORCIDs

James A Watson (ID) https://orcid.org/0000-0001-5524-0325
Stije Leopold (ID) http://orcid.org/0000-0002-0482-5689
Hugh Kingston (ID) http://orcid.org/0000-0003-1869-8307
Kathryn Maitland (ID) http://orcid.org/0000-0002-0007-0645
Nicholas PJ Day (ID) http://orcid.org/0000-0003-2309-1171
Arjen M Dondorp (ID) http://orcid.org/0000-0001-5190-2395
Thomas N Williams (ID) https://orcid.org/0000-0003-4456-2382
Nicholas J White (ID) http://orcid.org/0000-0002-1897-1978

## Ethics

Human subjects: All clinical data are from published studies in which all participants or guardians gave fully informed consent. Access to the human genetic data was approved by the MalariaGen data access committee.

## Decision letter and Author response

Decision letter https://doi.org/10.7554/eLife.69698.sa1
Author response https://doi.org/10.7554/eLife.69698.sa2

# Additional files

## Supplementary files

• Transparent reporting form

## Data availability

A curated minimal clinical dataset is currently available alongisde the code on the github repository. This is also available via the KEMRI-Wellcome Harvard Dataverse (https://doi.org/10.7910/DVN/TH8WAW). Whole genome data are available from European Genome-Phenome Archive (dataset accession ID: EGAD00010001742). Requests for access to appropriately anonymized clinical data and directly typed genetic variants for the Kenyan severe malaria cohort can be made by application to the data access committee at the KEMRI-Wellcome Trust Research Programme by e-mail to mmunene@kemri-wellcome.org. The FEAST trial datasets are available from the principal investigator on reasonable request (k.maitland@imperial.ac.uk). Requests for access to appropriately anonymized clinical data from the AQ and AAV Vietnam study and the Asian paediatric cohort can be made via the Mahidol Oxford Tropical Medicine Research Unit data access committee by emailing the corresponding author JAW (jwatowatson@gmail.com) or Rita Chanviriyavuth (rita@tropmedres.ac).

The following dataset was generated:

| Author(s) | Year | Dataset title | Dataset URL | Database and Identifier |
|---|---|---|---|---|
| Watson JA, Maitland K, Williams TN, White NJ | 2021 | Replication Data for: Improving statistical power in severe malaria genetic association studies by augmenting phenotypic precision | https://doi.org/10.7910/DVN/TH8WAW | Harvard Dataverse, 10.7910/DVN/TH8WAW |

The following previously published dataset was used:

| Author(s) | Year | Dataset title | Dataset URL | Database and Identifier |
|---|---|---|---|---|
| MalariaGen Consortium | 2015 | A genome-wide study of resistance to severe malaria in 18,000 samples from eleven worldwide populations, including eight populations sub-Saharan Africa. | https://ega-archive.org/studies/EGAS00001001311 | European Genome-Phenome Archive, EGAD00010001742 |

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

## Appendix 1

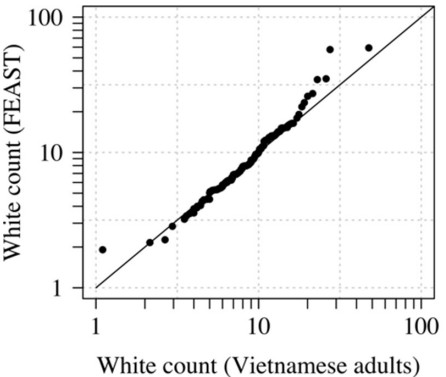
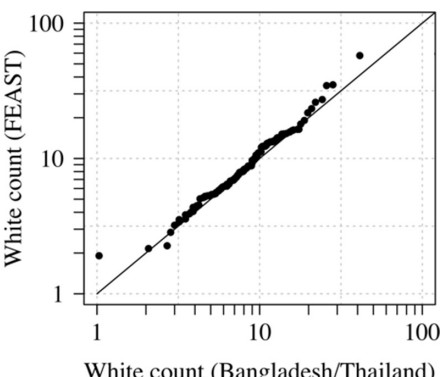

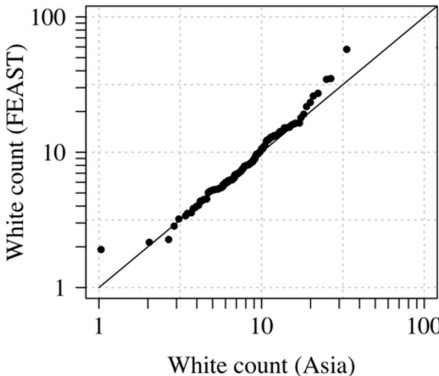

**Appendix 1—figure 1.** Comparison of the marginal distributions of white blood cell counts between Asian adults and children with severe malaria and African children with severe malaria. FEAST: 121 severely ill Ugandan children with *Pf*HRP2 >1000 ng/mL (***Maitland et al., 2011***). Vietnamese adults: 930 adults from two large randomised trials in severe malaria (***Phu et al., 2010***; ***Hien et al., 1996***). Bangladesh/Thailand: 653 adults and children from observational studies of severe malaria (***Leopold et al., 2019***).

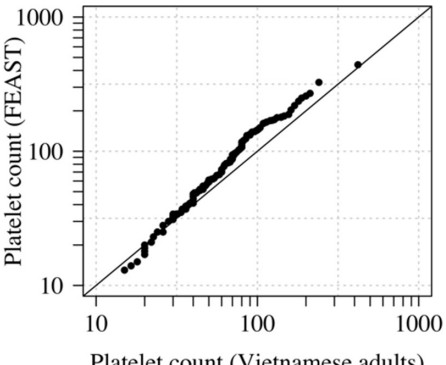
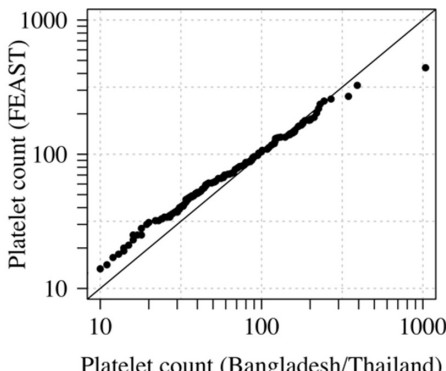

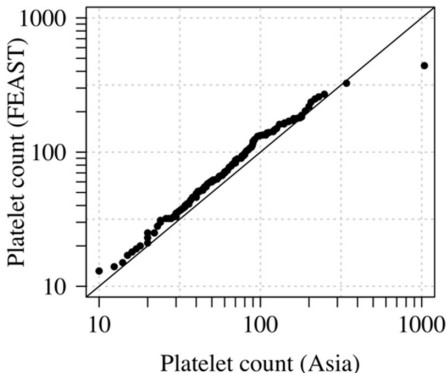

**Appendix 1—figure 2.** Comparison of the marginal distributions of platelet counts between Asian adults and children with severe malaria and African children with severe malaria. FEAST: 121 severely ill Ugandan children with *Pf*HRP2 >1000 ng/mL (*Maitland et al., 2011*). Vietnamese adults: 930 adults from two large randomised trials in severe malaria (*Phu et al., 2010*; *Hien et al., 1996*). Bangladesh/Thailand: 653 adults and children from observational studies of severe malaria (*Leopold et al., 2019*). The bottom-left qqplot compares the white counts from the children in the FEAST study with the combined dataset from Vietnam and Bangladesh/Thailand.

## Appendix 2

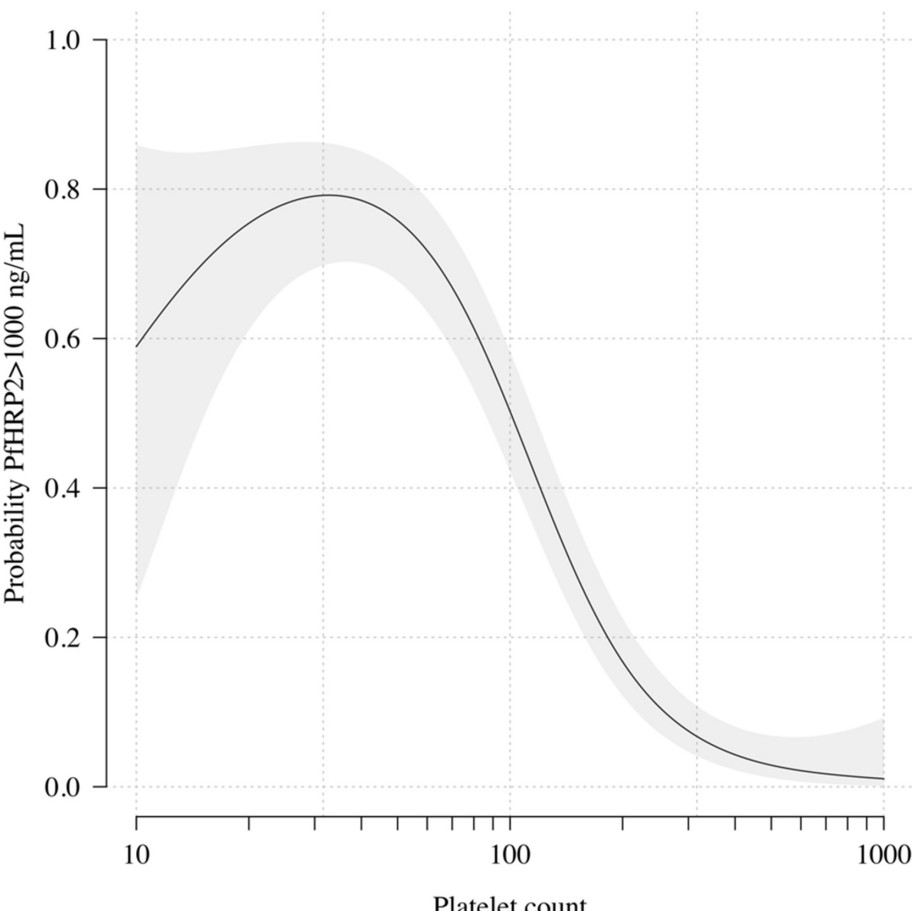

**Appendix 2—figure 1.** The relationship between platelet counts and plasma *Pf*HRP2 in severely ill African children. The black line (shaded area) shows the estimated probability (95% confidence interval) that the plasma *Pf*HRP2 >1000 ng/mL as a function of $log_{10}$ platelet count. This fit is derived from a generalised additive logistic regression model ($p<10^{-16}$ for the spline term), fit using the R package *mgcv*. The generalised additive model was fit to data from 566 African children enrolled in the FEAST trial (*Maitland et al., 2011*) (all the children who had both platelet counts and *Pf*HRP2 data available). Plasma *Pf*HRP2 >1000 ng/mL is highly discriminatory for severe malaria (*Hendriksen et al., 2012*).

## Appendix 3

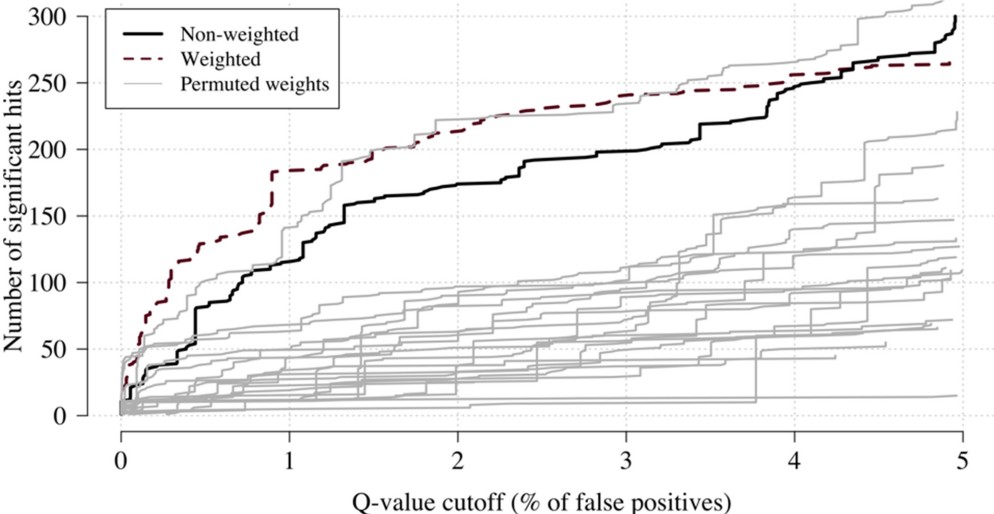

**Appendix 3—figure 1.** Effect of permuting the weights in the re-weighted (data-tilting) GWAS. Here we show the results of 20 random permutations of the weights, applied to the Kenyan case-control GWAS using only chromosomes 4, 9 and 11 (where the top hits are – we limit it to these three chromosomes for computational reasons). The random permutations (grey) decrease the number of significant hits compared to the non-weighted (thick black) and the non-permuted re-weighted model (dashed purple).

## Appendix 4

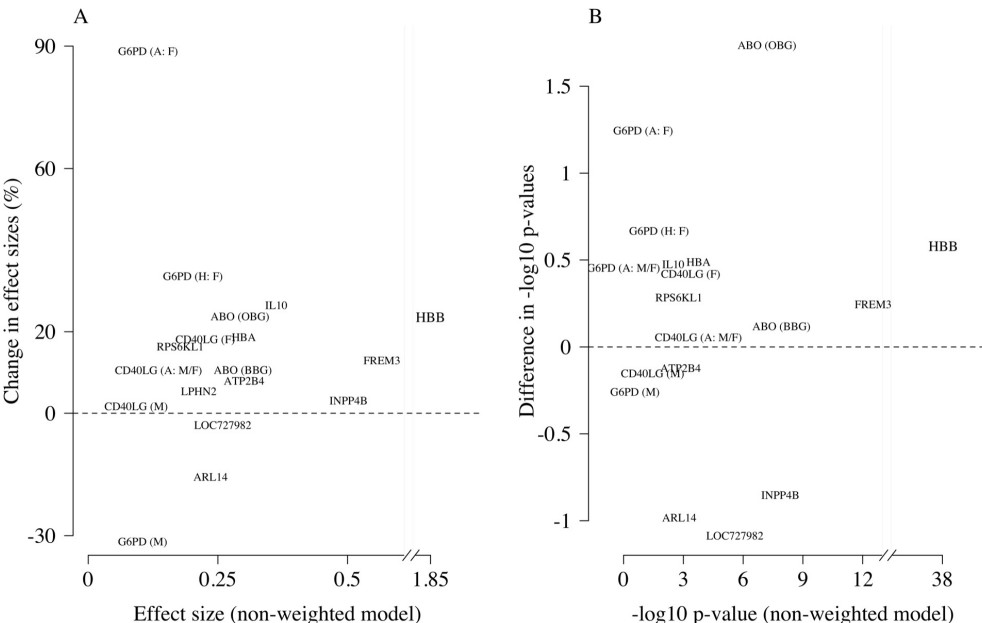

**Appendix 4—figure 1.** Comparison of the non-weighted and weighted models of association for directly typed polymorphisms previously reported as associated with severe malaria (*MalariaGEN Consortium et al., 2018*). (**A**) Estimated effect sizes under the non-weighted model versus the difference in effect sizes between the weighted and non-weighted models (absolute effects on the log-odds scale). Differences > 0 imply that the absolute effect size is estimated to be larger under the weighted model. (**B**) $-log_{10}$ p-values under the non-weighted model versus the differences in $-log_{10}$ p-values under the weighted and non-weighted models, again differences > 0 represent larger $-log_{10}$ p-values for the weighted model. Each point is represented by the gene name. In each case, we use the model that best fit the data in the original analysis (*MalariaGEN Consortium et al., 2018*). For the X-linked polymorphisms (*G6PD, CD40LG*), multiple models were reported and so the association model is also shown. H: heterozygote; A: additive; M: males only; F: females only; M/F: all.

## Appendix 5

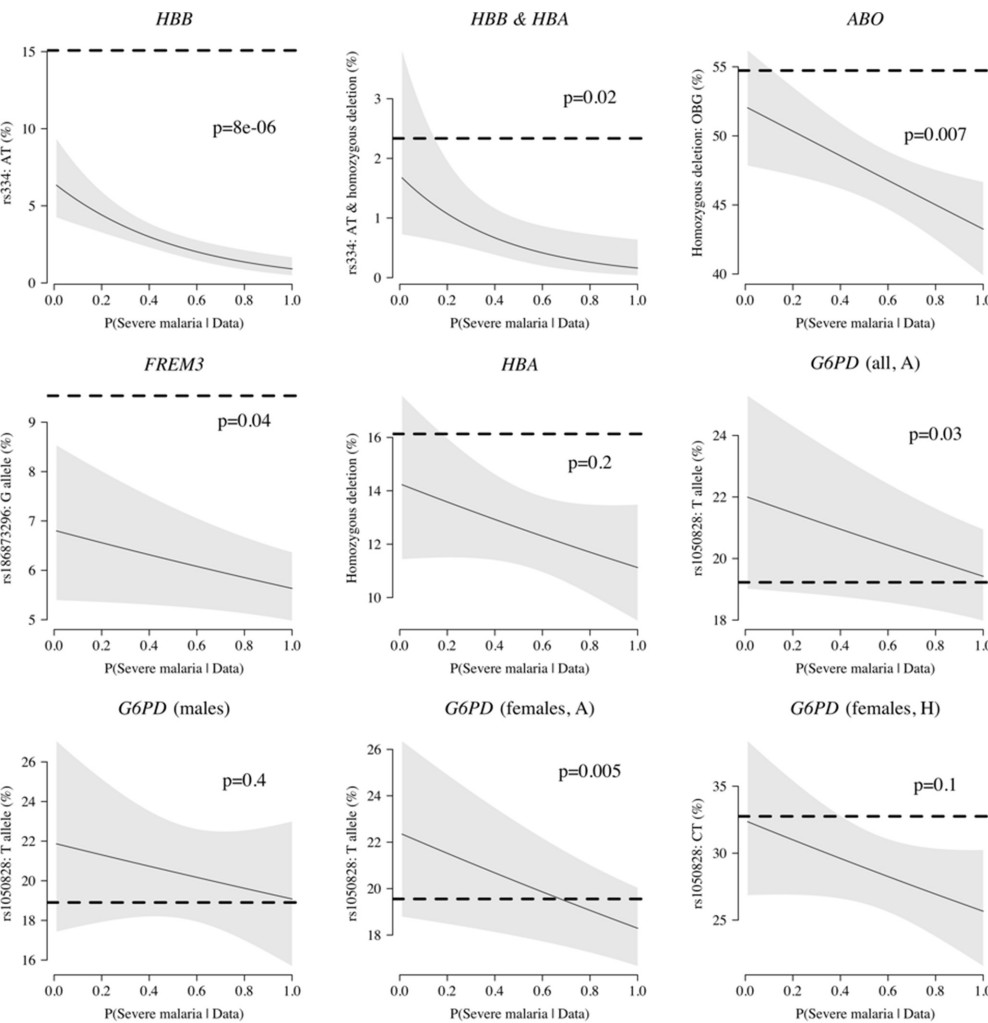

**Appendix 5—figure 1.** Case-only analysis of five key polymorphisms effecting red cells, reported in *Ndila et al., 2020* under additive, recessive or heterozygote models. The horizontal dashed lines show the estimated frequency in the controls (for additive models, this is the frequency of the derived allele; for the heterozygote or recessive models, this is the frequency of the genotype thought to confer protection). The line (shaded area) shows logistic regression fits with P(Severe malaria | Data) as the predictor (95% confidence interval of the fit). The p-value corresponds to the test that the predictor P(Severe malaria | Data) is not associated with the genotype in the cases only. OBG: O blood group.

## Appendix 6

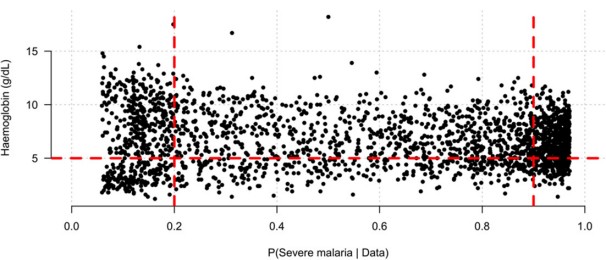

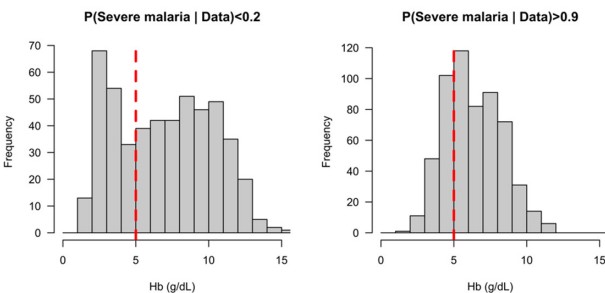

**Appendix 6—figure 1.** Distribution of admission haemoglobin concentrations as a function of P (Severe malaria | Data). Severe anaemia is generally defined as a haemoglobin less than 5 g/dL in African children diagnosed with severe malaria, shown by the horizontal dashed red line in the top panel and the vertical dashed red lines in the bottom panels. The vertical dashed red lines in the top panel show the top and bottom quintiles of the probability distribution (0.9 and 0.2, respectively). Patients in the bottom quintile of the probability distribution had a markedly bimodal distribution in haemoglobin concentrations with a substantial proportion meeting the severe anaemia criterion and a substantial proportion with relatively high haemoglobin concentrations (>10 g/dL), suggesting two patients subgroups. Patients in the top quintile had a unimodal distribution of haemoglobin.

## Appendix 7

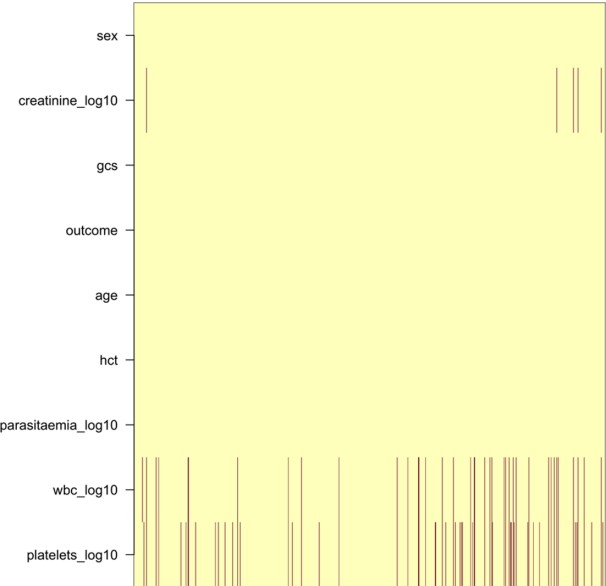

**Appendix 7—figure 1.** Pattern of missing clinical data in the 930 Vietnamese adults. These data pool the AQ Vietnam severe malaria study (*Hien et al., 1996*) and the AAV severe malaria study (*Phu et al., 2010*) (red: missing; yellow: recorded).

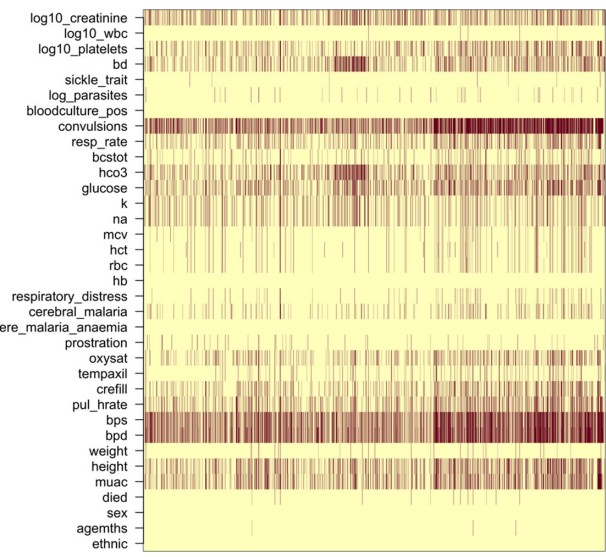

**Appendix 7—figure 2.** Missing clinical data in the 2220 Kenyan children diagnosed with severe malaria (red: missing; yellow: recorded).

## Appendix 8

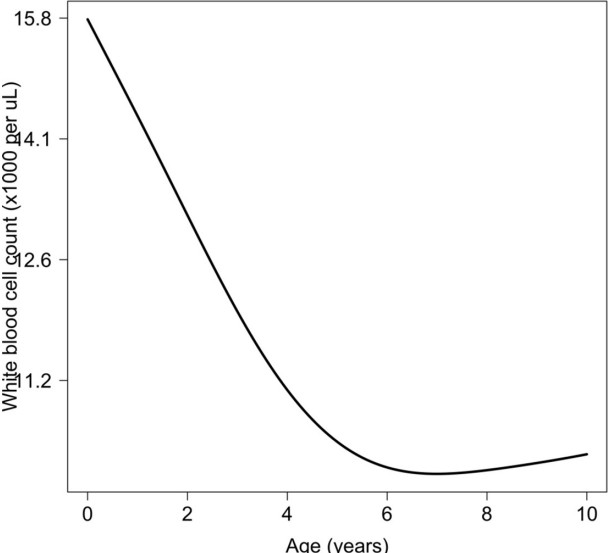

**Appendix 8—figure 1.** Relationship between age and mean white count (modelled on the $log_{10}$ scale). This is estimated from 858 children in the FEAST trial who had white counts available using an additive linear model ($p = 10^{-8}$ for the smooth spline term). We used this model to adjust observed $log_{10}$ white counts in all children less than 5 years of age in the reference and Kenyan datasets.

## Appendix 9

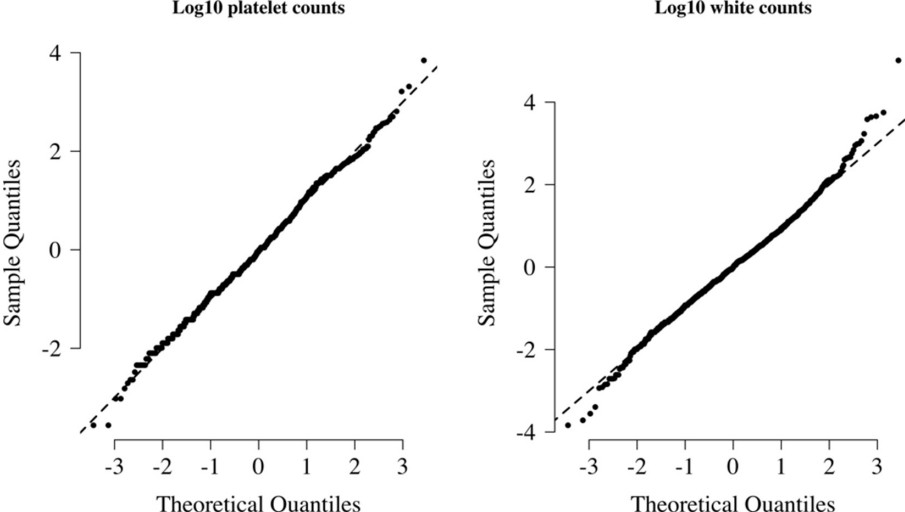

**Appendix 9—figure 1.** Normal-quantile plots for platelet counts and white blood cell counts in the reference data. Both were standardised to have mean 0 and standard deviation of 1 on the $log_{10}$ scale. The diagonal lines show the identity line.

**Appendix 10**

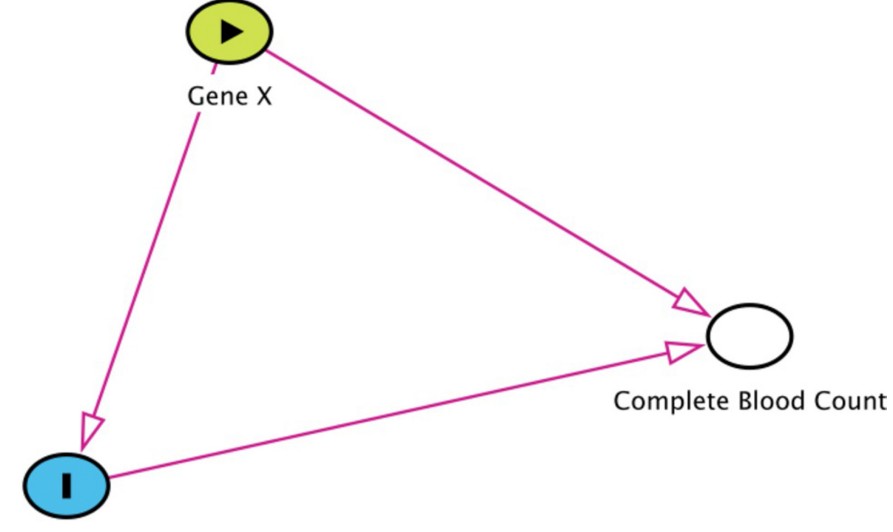

**Appendix 10—figure 1.** Collider bias in the diagnostic model of severe malaria based on complete blood count data. *HBB* in its homozygous S form (HbSS, <1% prevalence in this Kenyan population) is a rare example of how this can occur. Children with HbSS have white counts above 2–3 times higher than the normal population and slightly lower platelet counts (*Sadarangani et al., 2009*). Under the probabilistic model, all 11 children with HbSS were classified as having a low probability of severe malaria, based on their high white counts (mean 40,000 per μL). These probabilities cannot be taken at face value, and it remains an unanswered question whether children with HbSS are more or less susceptible than their wild-type counterparts (*Williams and Obaro, 2011*).

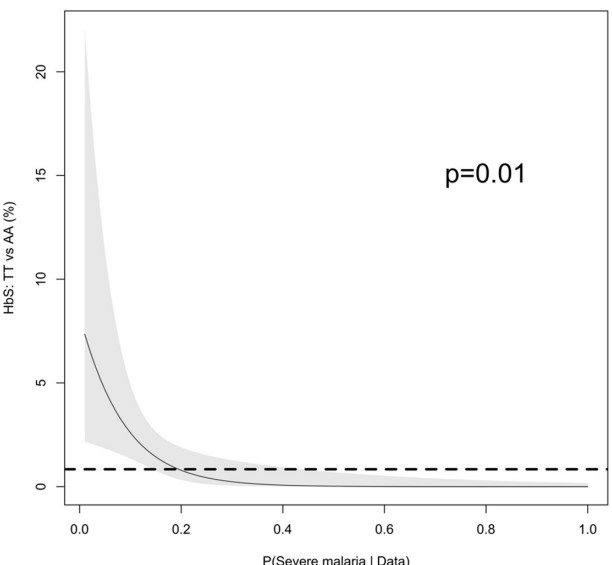

**Appendix 10—figure 2.** The relationship between HbSS and the estimated probabilities of severe malaria under the diagnostic model. There were 11 children with HbSS and they all had low probabilities of severe malaria, but this is biased as these children have chronic inflammation with
*Appendix 10—figure 2 continued on next page*

white counts 2–3 higher than the general population (*Sadarangani et al., 2009*) (see above *Appendix 10—figure 1* for the causal diagram showing collider bias).

## Appendix 11

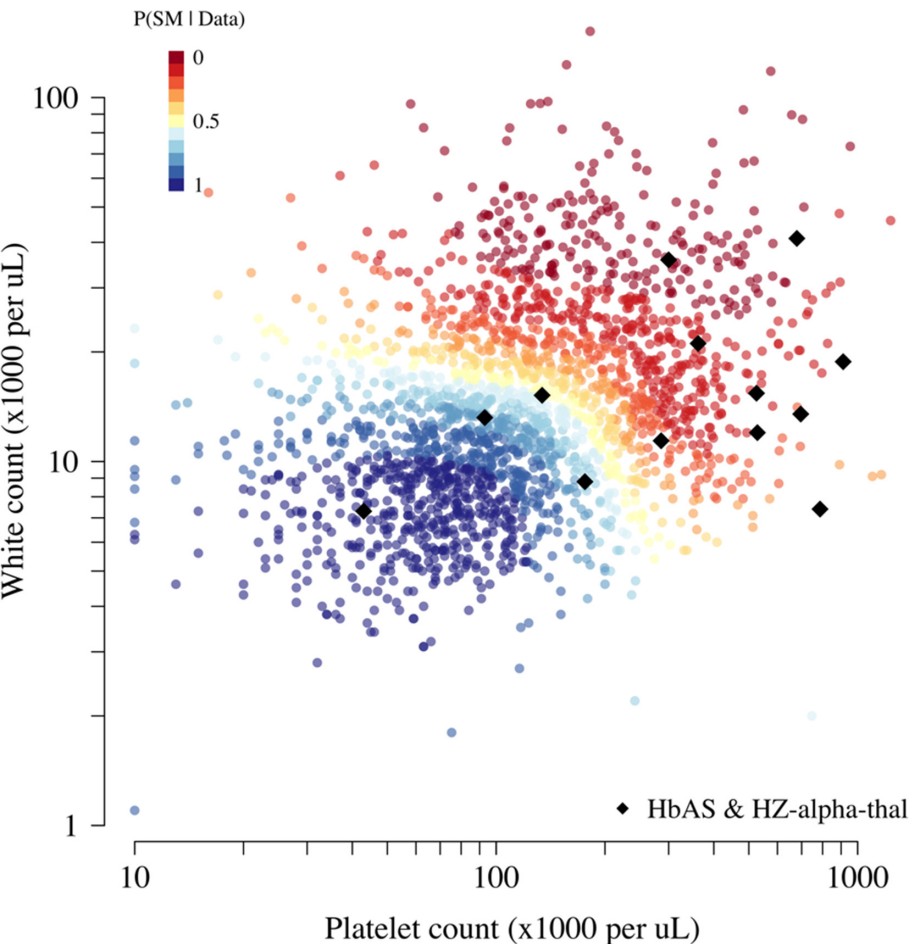

**Appendix 11—figure 1.** Scatter plots of platelet counts versus white blood cell counts for the Kenyan cohort, showing the 13 individuals with the double mutation HbAS and homozygous $\alpha^+$-thalassaemia as large black diamonds (HZ-alpha-thal). The red-yellow-blue colour scheme is proportional to the P(Severe malaria | Data) as given by the legend in the top-left corner.

## Appendix 12

### Simulation study

To demonstrate how the re-weighted likelihood works on simulated data where the true latent classes are known, we constructed the following simulation assuming

- A biallelic marker with a derived allele frequency of 10% in the control population (diplotypes encoded as 0, 1, 2).
- An additive protective effect for the true cases resulting in a derived allele frequency of 7% in the true cases; no effect in the false cases.
- The latent class probability weights for the true cases are drawn from a Beta(0.2, 1) distribution, and the probability weights for the false cases are drawn from a Beta(1, 0.2) distribution.
- A proportion of true versus false cases varying between 50% and 100%.

The R code for the simulation is given in the file Simulation_study_weightedLikelihood.R in the GitHub repository https://github.com/jwatowatson/Kenyan_phenotypic_accuracy. *Figures 1* and *2* show how the estimates effect sizes, the standard errors and the power (1-type 2 error) vary as a function of the proportion of the true cases.

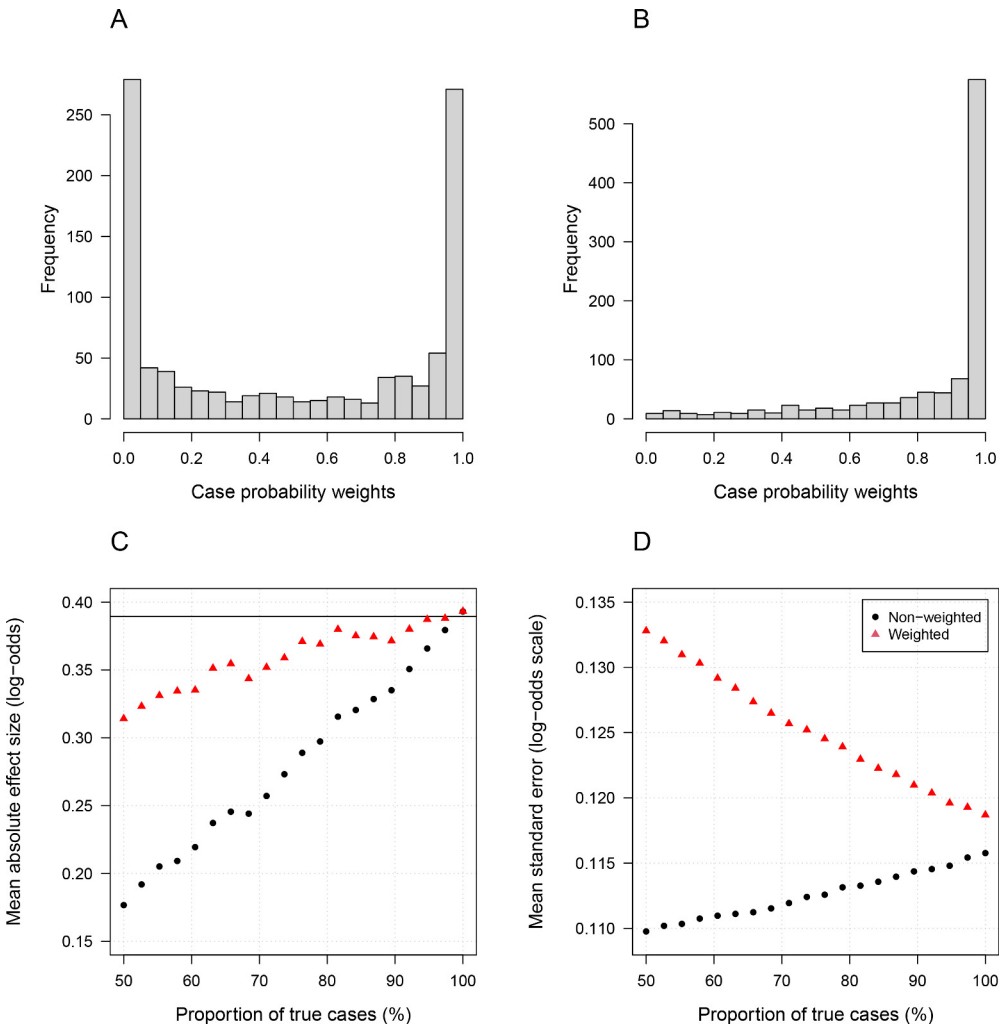

**Appendix 12—figure 1.** Simulation study demonstrating how likelihood re-weighting can improve estimation accuracy in case-control studies. Panels (**A**) and (**B**) show histograms of the case probability weights used in the simulations for the scenarios when 50% of cases are true cases and when 100% of cases are true cases, respectively. Panel (**C**) shows the estimated effect sizes as a function of the proportion of mis-classified cases. Panel (**D**) shows the standard errors of effect estimates as a proportion of mis-classified cases.

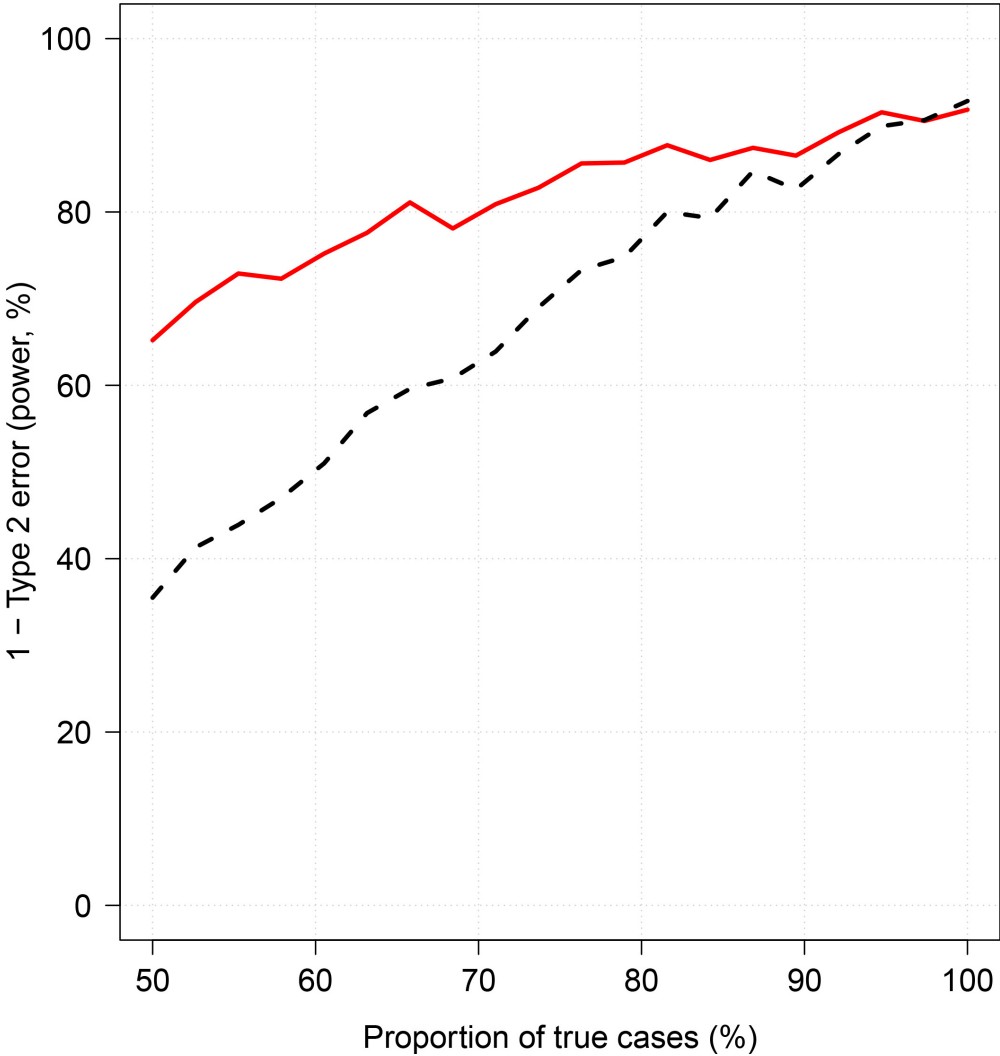

**Appendix 12—figure 2.** Effect of case re-weighting on power (1-type 2 error). The thick red line shows the estimated power for the re-weighted approach; the dashed black line shows the estimated power for the non-weighted approach.

## Appendix 13

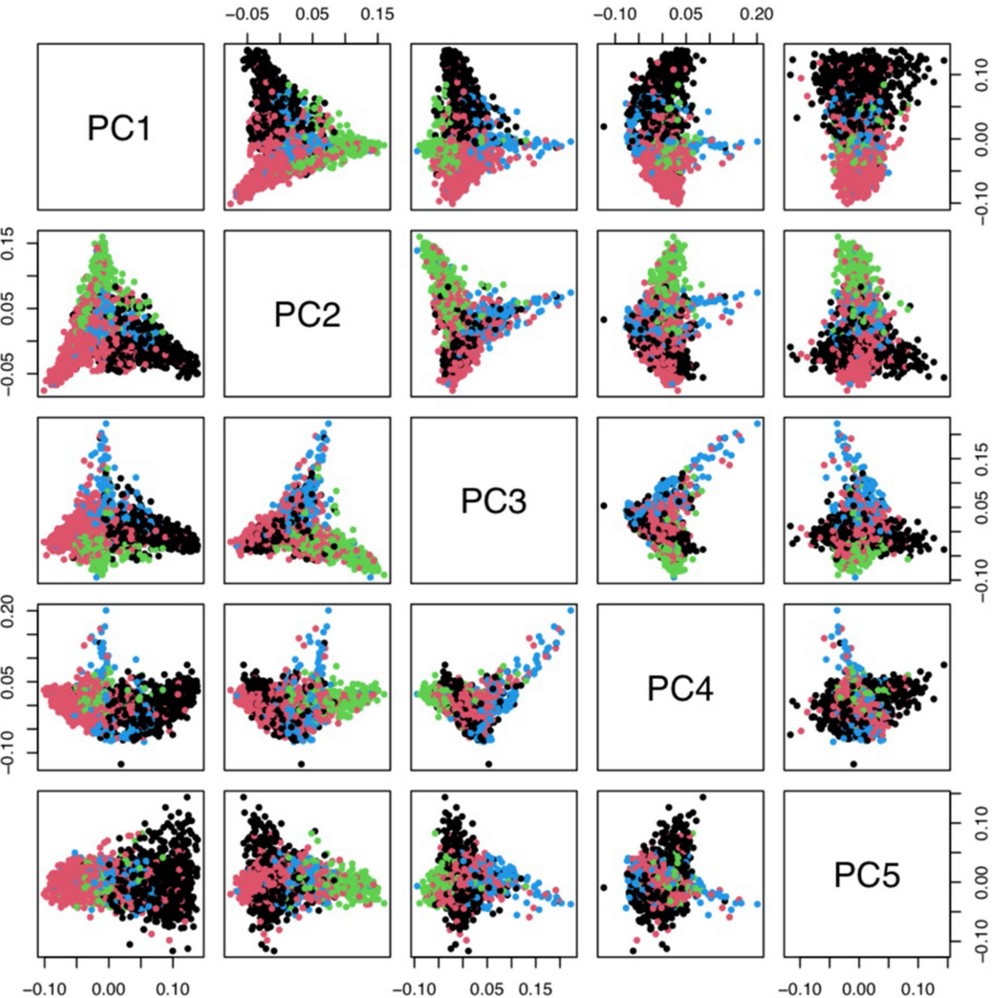

**Appendix 13—figure 1.** Principal components analysis of 1666 Kenyan cases and 1606 population controls. The colours show the main self-reported ethnicities (black: Chonyi; red: Giriama; green: Kauma; blue: other). The first five principal components were used to stratify for population structure in the GWAS analyses.

