## [Decision Letter]

**Acceptance summary:**

The fundamental premise of genome wide association studies for severe malaria is to take a population with confirmed severe malaria and compare with a control group who do not have severe malaria. This paper presents a novel and valuable method for improving power for severe malaria genetic association studies. The method would also be useful for studies of other disease where there is a clinical definition that sometimes includes people who do not truly have the disease.

**Decision letter after peer review:**

Thank you for submitting your article "Improving statistical power in severe malaria genetic association studies by augmenting phenotypic precision" for consideration by *eLife*. Your article has been reviewed by 2 peer reviewers, and the evaluation has been overseen by a Reviewing Editor and Dominique Soldati-Favre as the Senior Editor. The following individual involved in review of your submission has agreed to reveal their identity: Michael White (Reviewer #2).

Essential Revisions:

1) The paper would benefit from a clear definition early on of what is meant by "true severe malaria" – is it severe malaria symptoms that would go away if the malaria parasites are removed or severe malaria symptoms that would persist if all non-malaria illness was removed.

2) Page 3. "To rule out confounding by geography and age, we demonstrate the discriminatory value of platelet counts alone." It wasn't clear why this ruled out confounding by geography and age until it was discussed several sentences later that "Total white blood cell counts are age dependent and vary across genetic backgrounds."

3) For the data tilting method, how does one interpret the estimated odds ratio?

Is it the effect of the genotype on true severe malaria?

4) A simulation study would be useful in understanding the statistical properties of the proposed method.

5) A table summarizing the cohorts with summary statistics such as geographic location, age, symptom severity, and other relevant epidemiological information would be very useful.

6) A more explicit demonstration that what we observe about severe malaria in Asian adults applies to Asian children, applies to African children.

7) Figure 1B. For the grey line fitted to the FEAST data, does this also include the PfHRP2 = 1 data. As this was non-detectable, is this a valid thing to do?

8) Figure 3. Can you check the panel labels? What's the horizontal dashed line?

9) Were they significant associations between parasite density and the probability of severe malaria.

10) Please provide some additional information on the comparison between the training and testing data used for the model.

Reviewer #1 (Recommendations for the authors):

The paper would benefit from a clear definition early on of what is meant by "true severe malaria" – is it severe malaria symptoms that would go away if the malaria parasites are removed or severe malaria symptoms that would persist if all non-malaria illness was removed.

Page 3. "To rule out confounding by geography and age, we demonstrate the discriminatory value of platelet counts alone." It wasn't clear to me why this ruled out confounding by geography and age until it was discussed several sentences later that "Total white blood cell counts are age dependent and vary across genetic backgrounds."

For the data tilting method, how does one interpret the estimated odds ratio?

Is it the effect of the genotype on true severe malaria?

This may not be needed for an *eLife* paper but simulation studies would be useful in understanding the statistical properties of the proposed method.

Reviewer #2 (Recommendations for the authors):

Please provide some additional information on the comparison between the training and testing data used for the model.

---

## [Author Response]

Essential Revisions:1) The paper would benefit from a clear definition early on of what is meant by "true severe malaria" – is it severe malaria symptoms that would go away if the malaria parasites are removed or severe malaria symptoms that would persist if all non-malaria illness was removed.

We agree that this is important and central to the exercise, and was left too vague in the initial submission.

We define “true severe malaria” conceptually as (i) a febrile illness caused by malaria parasites, (ii) with an elevated risk of death, (iii) and in which mortality is directly attributable to the malaria parasites.

The two conditions which we are trying to discriminate between are severe illness caused by malaria and severe illness with coincidental malaria. A more formal definition can be given using a thought experiment: an idealized (but unethical) randomized trial whereby antimalarial treatment is delayed vs given promptly. In a population of “true severe malaria” patients, delay in administration of an effective antimalarial would result directly in increased mortality, whereas this would not be the case in “not severe malaria”. In clinically diagnosed severe malaria delay in starting curative treatment is associated with an increased mortality (Warrell et al. NEJM 1982), and pre-referral treatment in suspected severe malaria has been shown to reduce mortality and morbidity (Gomes et al. Lancet 2009). The life-saving advantage of artesunate over quinine in the large RCTs AQUAMAT and SEAQUAMAT can also be interpreted as supporting this in that artesunate kills circulating ring stage parasites whereas quinine does not, resulting in a delay in the onset of therapeutic benefit, thus mimicking a “delay” in treatment. There was evidence in the AQUAMAT trial of an increased treatment effect as a function of the admission plasma *Pf*HRP2 concentration, consistent with the hypothesis that the subgroup of patients with high *Pf*HRP2 has higher specificity for “true severe malaria” (Hendriksen et al. PLoS Med 2012).

We have added the following paragraph to the penultimate paragraph in the Introduction (lines 93-104):

Our goal was to develop a biomarker-based model that can differentiate probabilistically between `true severe malaria' and severe illness not caused primarily by malaria, but with concomitant parasitaemia. We define `true severe malaria' conceptually as a febrile illness caused by malaria parasites, with organ dysfunction, that can result in death whereby mortality is attributable directly to the malaria parasites. This attributable mortality can be given a formal causal definition by using a conceptual (albeit unethical) randomised experiment of delayed versus prompt anti-malarial therapy. In a theoretical patient population with true severe malaria, delay in administration of an effective antimalarial would result in increased mortality, whereas in a population with severe illness not caused by malaria (`not severe malaria') there would not be a corresponding increase in mortality.

2) Page 3. "To rule out confounding by geography and age, we demonstrate the discriminatory value of platelet counts alone." It wasn't clear why this ruled out confounding by geography and age until it was discussed several sentences later that "Total white blood cell counts are age dependent and vary across genetic backgrounds."

This paragraph was unclear and we have rewritten it. We have removed the “rule out” as it is not possible in this context to rule out confounding. The re-written paragraph now reads as (lines 120-137):

Direct comparisons of white blood cell counts across these two data sets are confounded by geography and age. Total white blood cell counts are known to be age-dependent and vary across genetic backgrounds, in particular lower neutrophil counts are associated with mutations in the ACKR1 gene that results in the Duffy negative phenotype prevalent in African populations (Reich et al., 2009). However, after adjustment for age (see Methods), the marginal distributions of total white counts were comparable between Asian adults and children with severe malaria and African children with high plasma PfHRP2 (Appendix 2). Platelet counts are not age dependent and do not vary substantially across genetic backgrounds. The marginal distributions of platelet counts were comparable between Asian adults and children with severe malaria and African children with high plasma PfHRP2 (Appendix 1). A low platelet count (thrombocytopenia) is a universal feature of severe malaria (see evidence collated in Methods). To illustrate this important point, in a cohort of 566 severely ill Ugandan children enrolled in the FEAST trial (Maitland et al., 2011, a trial including children with severe febrile illness, not restricted to severe malaria), low platelet counts were highly predictive of blood stage parasitaemia and elevated PfHRP2 (p=10^-16^ for a spline term on the log_10_ platelet count in a generalised additive logistic regression model predicting PfHRP2 > 1,000 ng/mL, Appendix 1). Children enrolled in the FEAST trial who had significant thrombocytopenia (<100,000 platelets per uL) had comparable PfHRP2 concentrations to Asian adults diagnosed with severe falciparum malaria (Figure 1B).

3) For the data tilting method, how does one interpret the estimated odds ratio?Is it the effect of the genotype on true severe malaria?

Yes, exactly. This is analogous to the interpretation of regression models when using inverse probability weighting (targeting the causal estimand for the pseudo population obtained through re-weighting). We have added the following sentence to the Methods (lines 545-547):

The log odds ratio computed from the weighted logistic regression can be interpreted as the causal effect of the polymorphism on `true severe malaria' relative to the controls, where `true severe malaria’ is defined by the sampling distribution f0.

4) A simulation study would be useful in understanding the statistical properties of the proposed method.

We have added a simple simulation study to the github repository which is reported in Appendix 12. The simulation assumes that cases are composed of two latent classes (true cases where the genotype of interest has a lower allele frequency; and false cases where the genotype of interest has the same allele frequency as the controls), for which we have calibrated class probabilities (ie the class probabilities are proportional to the latent class). We then varied the proportion of the true cases from 50% to 100%. Figure 1 in Appendix 12 shows the distribution of the case weights for the 50% contamination and 0% contamination scenarios (panels A and B), and shows how the effect estimates and standard errors vary as a function of the proportion of true cases (panels C and D). When we use the information in the case probability weights (weighted log-likelihood) the mean effect estimates on the log odds scale (shown by the red triangles, averaged over 1000 simulations) are close to the true simulated effect for 50% contamination and recover exactly the true effect when the proportion of true cases > 80%. This increase in precision in effect estimation comes at the cost of an increase in the standard errors (due to the reduced effective sample size). Figure 2 in Appendix 12 shows the resulting effect on statistical power. It is important to note that the advantage of the re-weighted likelihood approach relies heavily on the calibration and precision of the case probability weights. If the diagnostic model is biased or only weakly informative (eg the majority of the weights are not close to 0 or 1) then likelihood re-weighting will not add much and could reduce power.

5) A table summarizing the cohorts with summary statistics such as geographic location, age, symptom severity, and other relevant epidemiological information would be very useful.

We have added Table 1 which summarizes the 4 data sets used in the analysis.

6) A more explicit demonstration that what we observe about severe malaria in Asian adults applies to Asian children, applies to African children.

The number of Asian children with severe malaria studied in our analysis is relatively small by comparison with the number of Asian adults. There is no evidence that the marginal distributions of platelet counts are different between the Asian adults and children or the Asian adults and the African children with high *Pf*HRP2. The same is true for the white counts after age adjustment. Although there are clinical differences between adults and children with severe malaria, the fundamental pathological processes appear to be the same. This is supported by the fact that the life-saving treatment advantage of artesunate over quinine is the same when comparing children with high plasma *Pf*HRP2 (i.e. specific for severe malaria) versus adults (see Hendriksen et al. PLoS Med, 2012).

We argue that the proposed reference model for severe malaria is justified for two following reasons. First, we make an age-adjustment to the white counts (although this is far from perfect); second, the relationship between platelet counts and *Pf*HRP2 in African children (FEAST trial) strongly supports low platelet counts as a universal feature of severe malaria

The model allows for varying mean platelet counts and white counts across the datasets: in the Kenyan children the sampling distribution P(Data | Severe Malaria) is modelled as a mixture of two components, one equal to the fit using Asian adults/children and the other equal to the fit using the 121 children in FEAST (these two distributions are modelled as being drawn from the same hierarchical distribution so that information is shared).

7) Figure 1B. For the grey line fitted to the FEAST data, does this also include the PfHRP2 = 1 data. As this was non-detectable, is this a valid thing to do?

The grey line (changed to brown in the revised paper so that it is more visible) does include the non-detectable *Pf*HRP2 concentrations which were set to 1 ng/mL. If we assume that all these patients did indeed have a PfHRP2 concentration less than the lower limit of the ELISA assay (approximately 2 ng/mL, ie assuming that these are not assay errors) then we believe that this is valid with some limitations on the interpretation of the regression fit. The analysis is using a log_10_(x+λ) transformation (a Box-Cox transformation). Choosing λ=1 (or half the lower limit of detection) is widely used, for example in pharmacometric analyses where drug concentrations are often below the lower limit of detection for a particular assay. This can introduce bias in parameter estimation for pharmacokinetic models (see for example https://www.ncbi.nlm.nih.gov/pmc/articles/PMC2691472/) but we can’t see any issue in this particular setting as we are only demonstrating the correlation between *Pf*HRP2 and platelet counts in severe illness rather than attempting to infer a population mean *Pf*HRP2 concentration for a given platelet count.

One issue could be for patient populations where *HRP2/3* deletions are common. This could dilute the association between platelet counts and *Pf*HRP2 plasma concentrations. HRP2/3 deletions were rare in Uganda at the time of the FEAST trial confirmed by the fact that very few patients had measurable parasitaemia >1000 uL with undetectable plasma *Pf*HRP2.

8) Figure 3. Can you check the panel labels? What's the horizontal dashed line?

We have corrected the legend of Figure 3 (panels C and D). The horizontal dashed line in panel D was showing the mean mortality in the cohort. We have added the analogous lines to panels C and to the new panel E (parasite density – as suggested in next point). We have included explanations in the legend.

9) Were they significant associations between parasite density and the probability of severe malaria.

Thank you. This is a very important question that we had addressed but then forgot to include in the paper! There is a significant relationship between the model probabilities and the admission parasite densities (*p*=10^-25^ for a linear model fit to log parasite densities). Patients with a probability of severe malaria approximately equal to 1 under our model have a mean parasite density of ~70,000 per uL, whereas those with a probability approximately equal to 0 have a mean parasite density of ~13,000 per uL (roughly a fivefold difference). We have added a plot of model probabilities versus parasite densities to Figure 3 (panel E in the revised manuscript). These additional results are now described on lines 175-177 and 184-186:

The individual probabilities were also predictive of in-hospital mortality (p=10^-9^ from a generalised additive model fit; Figure 3D) and admission peripheral blood parasite density (p=10^-25^ from a generalised additive model fit; Figure 3E). [..] The admission parasite densities in patients with a probability of severe malaria close to 1 were approximately five-fold higher than in patients with a probability of severe malaria close to zero.

10) Please provide some additional information on the comparison between the training and testing data used for the model.

See Table 1 and replies to comment 6.